# Global soil antibiotic resistance genes are associated with increasing risk and connectivity to human resistome

Yuxiang Zhao[1], Liguan Li[1], Yue Huang [1], Xiaoqing Xu[1], Zishu Liu [2], Shuxian Li[1], Lizhong Zhu[2], Baolan Hu [2,3] ✉ & Tong Zhang [1,4,5,6,7] ✉

Soil is a reservoir of antibiotic resistance genes (ARGs), and understanding its connection to human antibiotic resistome is crucial for the One Health framework. Rank I ARGs appear key to deciphering this relationship, but their global distribution and attribution in soil remain unclear. To fill this gap, we analyze 3965 metagenomic data (12 habitats, including soil, feces, sewage) and 8388 genomes of *Escherichia coli* isolates. Results show that soil ARG risk has increased over time (from 2008 to 2021). We introduce a "connectivity" metric that evaluates cross-habitat ARGs connectivity through sequence similarity and phylogenetic analysis, and reveal higher genetic overlap with clinical *E. coli* genomes (1985–2023) over time suggesting an increasing link between soil and human resistome. A comparison of 45 million genome pairs suggests that cross-habitat horizontal gene transfer (HGT) is crucial for the connectivity of ARGs between humans and soil. Finally, we compile clinical antibiotic resistance datasets (covering 126 countries from 1998 to 2022) and find significant correlations between soil ARG risk, potential HGT events and clinical antibiotic resistance ($R^2 = 0.40$–$0.89$, $p < 0.001$). Overall, our work provides insights into the ARGs connectivity between soil and humans, and could help identify strategies to prevent dissemination of antibiotic resistance.

Antimicrobial resistance (AMR) poses a growing threat to global health[1]. According to estimates from the Institute for Health Metrics and Evaluation, bacterial AMR was directly responsible for over 1.27 million human deaths[2]. By 2050, AMR will be responsible for 10 million deaths annually and result in a loss of $100 trillion from global gross domestic product[3]. Consequently, the AMR pandemic has been, and will continue to be, a critical challenge that beyond its effects on human health[4]. There is growing evidence that the origins of AMR may extend beyond human habitats[5,6]. Some antibiotic resistance genes (ARGs) originated from environmental bacteria[7] and were transferred to humans via direct or indirect contact, such as through food chains[8].

Soil is one of the main sources of microbes in terrestrial ecosystems[9] and acts as the cornerstone of One Health framework[10]. Consequently, increasing attention is now being paid to the biogeography of the soil antibiotic resistome. The global distribution of the abundance of ARGs (via high-throughput quantitative PCR)[11], the relative abundance of ARGs (via metagenomic analysis)[12], and the diversity of soil ARG-carrying pathogens (via metagenomic analysis)[13] have been mapped. Meanwhile, environmental factors[14,15] and climatic

[1]Environmental Microbiome Engineering and Biotechnology Laboratory, Department of Civil Engineering, The University of Hong Kong, Hong Kong SAR, China. [2]College of Environmental and Resource Sciences, Zhejiang University, Hangzhou, China. [3]Zhejiang Province Key Laboratory for Water Pollution Control and Environmental Safety, Hangzhou, China. [4]School of Public Health, The University of Hong Kong, Hong Kong SAR, China. [5]Center for Environmental Engineering Research, The University of Hong Kong, Hong Kong SAR, China. [6]Shenzhen Innovation and Research Institute, The University of Hong Kong, Shenzhen, China. [7]The State Key Laboratory of Marine Environmental Health (SKLMEH), City University of Hong Kong, Hong Kong SAR, China. ✉e-mail: blhu@zju.edu.cn; zhangt@hku.hk

seasonality[11] have been regarded as the main drivers shaping soil antibiotic resistome. While the distribution and drivers of ARGs in soil seem clear, the sources of ARGs in soil and their relationship to clinical antibiotic resistance within the One Health framework remain unclear.

Indeed, it is not surprising that ARGs are present in natural soils as they are ubiquitous[16] and can even be found in the Mariana Trench[17], undisturbed Alaskan soil[18,19], and 30,000-year-old permafrost sediments[20]. Multidrug efflux pumps are one of the main types of ARGs in soil[14]. However, it must be recognized that high abundance may not mean high risk and that only human-associated ARGs pose a serious health risk[21]. The One Health framework for antibiotic resistance focuses on ARGs that can cross ecological boundaries[8]. Therefore, identifying these "high-risk" ARGs from the presumptive ARGs is crucial for addressing this global issue with cost-effective strategies[22]. Zhang et al. developed an "omics-based" framework to evaluate ARG risk and identified the list of Rank I ARGs[23]. These Rank I ARGs were proposed based on host pathogenicity, gene mobility and human-associated enrichment, which are the focus of One Health[8]. Although Rank I ARGs are present in various habitats[24,25], their global soil distribution, attribution, and connectivity between soil and other habitats remain unclear. This knowledge gap limits not only identifying priority areas for combating the soil antibiotic resistome, but also understanding of the relevant health risks.

Herein, we constructed a dataset that includes 3816 metagenomic datasets (including 2391 soil samples and 1425 other habitat samples) from public databases and 149 in-house data to catalog the profile and attribution of the soil antibiotic resistome (i.e., 3965 samples in summary). Meanwhile, we collected 8388 *E. coli* genomes (the main ARGs carrying pathogen in soil) isolated from soil, livestock, and humans (main source), to reveal the potential exchange network of soil and other habitats Rank I ARGs at the species level. Finally, we constructed global human clinical antibiotic resistance datasets to examine the relationship of antibiotic resistance between the soil and humans. We aimed to address two questions: (i) What are the sources of Rank I ARGs, and how do they change over time? (ii) How does the soil antibiotic resistome relate to the human antibiotic resistome and to human clinical antibiotic resistance?

## Results
### The attribution and profile for global soil antibiotic resistome
To gain a comprehensive insight into the role of the soil antibiotic resistome within the One Health framework, we compiled a dataset of 3965 metagenomic samples (including 2540 soil samples and 1425 other habitat samples) (Figs. S1 and S2, and Supplementary Data 1–2). We analyzed the ARGs using ARGs-OAP (v 3.2.2) and used the relative abundance of Rank I ARGs to assess the risk of ARGs (genes listed in Supplementary Data 3 and Text S2)[26]. The profile of Rank I ARGs was obtained based on the previously defined list, characterized by reported host pathogenicity, gene mobility, and enrichment in human-associated environments[26]. In detail, after obtained the ARGs profile for 3965 metagenomic samples, we compared them to a previously established Rank I ARGs list to identify the presence of specific variants within these samples. The Rank I ARGs profiles represented the observation of these ARGs across the detected metagenomic samples in this study, while the classification of Rank I ARGs is based on the previous reports (Supplementary Data 3)[26]. The SARG3.0_S database, which excluded sequences associated with transcriptional regulators (including activators and repressors), point mutations, and others, was used for similarity search annotation[26]. In addition, considering the controversy surrounding multidrug efflux pumps, all genes related to these pumps were excluded to avoid possible mis-annotations of ARGs. The profiles of antibiotic resistome in different habitats and different land use types (including rarefaction curves and composition of ARGs at the type and subtype level) were shown in Figs. S3 and S6. Briefly, the richness (number of subtype) and relative abundance of total ARGs (1739 subtypes, 0.13 copies per cell) and Rank I ARGs

(175 gene types, 1.5 copies per 1000 cells) in soil were similar to those in wastewater treatment plant (WWTP) effluent but lower than those in livestock and human feces (Fig. 1a, b). The similar trends could also be observed after the minimum sample size for each habitat is standardized (Fig. S7). Principal Co-ordinates Analysis (PCoA) clearly separated the soil antibiotic resistome from those in other habitats (Adonis analysis, $p < 0.01$, Fig. 1c, d, Text S3, and Supplementary Data 4 and 5).

However, some significant differences were observed between total ARGs and Rank I ARGs in soil. The PCoA results for Rank I ARGs showed less intergroup dispersion compared to total ARGs, implying greater similarity in Rank I ARGs across soil samples (Fig. 1c, d). The trends of total ARGs and Rank I ARGs over time also showed significant differences (Fig. 1e, f). The relative abundance of total ARGs was time-independent ($r = 0.08$, $p > 0.05$, Fig. 1e). In contrast, both the relative abundance of Rank I ARGs ($r = 0.89$, $p < 0.001$) and their occurrence frequency (presence of Rank I ARGs samples / total number of samples for the corresponding year) ($r = 0.83$, $p < 0.001$) showed significant increases over time (Fig. 1f). To eliminate the effects of sample size and collection site across different years on the observed trends, we divided the data into five distinct periods (details in Methods). To further ensure the reliability of the trends, we performed data normalization on both year (Fig. S8) and period (Fig. S9a). The changes in the relative abundance of total ARGs and Rank I ARGs were assessed under consistent data volume (Fig. S9b), consistent continental origin combinations (Fig. S9c), and consistent land use type combinations (Fig. S9d). All these results were consistent with previous findings, indicating that the relative abundance of total ARGs was time-independent, but ARGs risk showed a significant increase over time. Furthermore, after representative ARGs (biomarker ARGs identified by LEfSe, Text S4, and Supplementary Data 6), we found no unique Rank I ARGs in soil. These results suggested that soil Rank I ARGs largely overlapped with ARGs in other habitats (i.e., human-related), and soil might be both the sink for Rank I ARGs.

To confirm this hypothesis, the fast expectation-maximization for microbial source tracking (FEAST) analysis was carried out to reveal the sharing of ARGs between soil and other habitats[27] (Fig. 1g). On average, soil shared 60.1% of total ARGs and 50.9% of Rank I ARGs with other habitats. Specifically, human feces (75.4%), chicken feces (68.3%), WWTP effluent (59.1%), and swine feces (53.9%) were attributed the most to the Rank I ARGs in soil. Moreover, regarding other habitats, soil shared 77.6%–91.9% of ARGs with crop surfaces, natural water, natural sediments, and marine water. After differentiating between different Rank I ARGs subtypes, we observed a consistent increase in *mph(A)*, *APH(3')-Ia*, *AAC(6')-le-APH(2'')-la*, *ANT(6)Ia*, *aadA*, *APH(6)-Id*, *aadA10*, *mef(B)*, and *APH(3'')-Ib*, along with the first detection of *NMD-19* in soil samples in 2021 (Fig. 1h). Thus, given the growing risk posed by Rank I ARGs in soil and their close relationship with those found in livestock and human feces, soil might be a key node for controlling the spread of ARGs under the One Health framework. Overall, more attention should be paid to Rank I ARGs in soil, as they may act as a sink/source for clinical ARGs in hotspot regions.

### Profile of pathogens carrying ARGs in soils
We annotated ARG-carrying contigs (assembled from the 2540 soil samples) and their taxonomy to reveal soilborne pathogens carrying ARGs (pARGs) and Rank I ARGs (pRank I). Only the prokaryotic pathogens listed in two published reference pathogen lists, the PHI-base (www.phi-base.org/) and Catalog of Human Transmitted Pathogenic Microorganisms 2023, were focused (Supplementary Data 7). Before further analysis, ARGs carried by plasmids were excluded using geNomad[28], and the ARGs carried by chromosomes were focused, because of the difficulty in identifying the host of plasmids. Firstly, we investigated the distribution of pARGs and pRank I. The results showed that pARGs were present in over 29.08% of the samples, and pRank I was detected in 5.57% of the samples (Fig. 2a). Among the collected

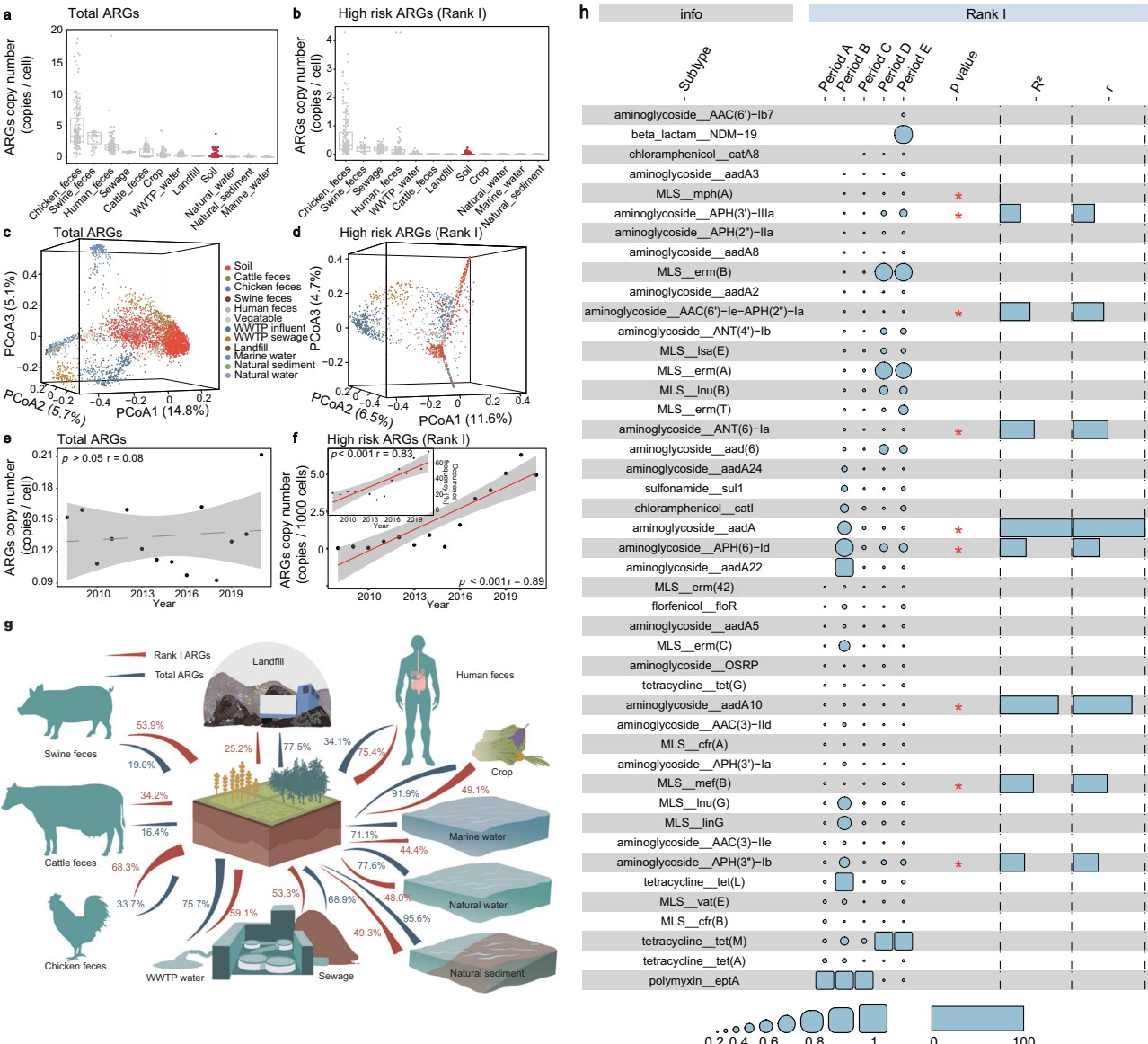

**Fig. 1 | The profile and attribution of global soil antibiotic resistome in metagenomic analysis. a** The relative abundance of total ARGs in various habitats. In the boxplots of panels, hinges indicate the 25th, 50th, and 75th percentiles, whiskers indicate 1.5× interquartile ranges, and dots indicate values of individual samples (Soil: 2540, WWTP_water: 303, Natural_sediment: 280, Marine_water: 188, Chicken_feces: 146, Human_feces: 130, Crop: 122, Sewage: 121, Natural_water: 70, Swine_feces: 31, Landfill: 22, Cattle_feces: 12). **b** The relative abundance of Rank I ARGs in various habitats. In the boxplots of panels, hinges indicate the 25th, 50th, and 75th percentiles, whiskers indicate 1.5× interquartile ranges, and dots indicate values of individual samples (Soil: 2540, WWTP_water: 303, Natural_sediment: 280, Marine_water: 188, Chicken_feces: 146, Human_feces: 130, Crop: 122, Sewage: 121, Natural_water: 70, Swine_feces: 31, Landfill: 22, Cattle_feces: 12). The classification of Rank I ARGs was based on a comparison with the previously reported list (Supplementary Data 3)[26]. **c** The PCoA with Bray-Curtis dissimilarity and Adonis analysis of total ARGs. A version with multiple viewing angles is available at https://github.com/Yuxiang-Zhao/ARGs/blob/main/PCoA/3D_PCoA_Total.html. The significance between different habitats could be observed in Supplementary Data 4. **d** The PCoA with Bray-Curtis dissimilarity and Adonis analysis of Rank I ARGs. A version with multiple viewing angles is available at https://github.com/Yuxiang-Zhao/ARGs/blob/main/PCoA/3D_PCoA_RankI.html. The significance between different habitats could be observed in Supplementary Data 5. The classification of Rank I ARGs was based on a comparison with the previously reported list (Supplementary Data 3)[26]. **e** The relative abundance of total ARGs over time. Each node represents the mean relative

abundance for that year. Gray shading denotes the 95% confidence intervals. Pearson correlation (liner regression) was conducted to compare the relationship between time and relative abundance of total ARGs. **f** The relative abundance of Rank I ARGs over time. Each node represents the mean relative abundance for that year. Gray shading denotes the 95% confidence intervals. Pearson correlation test (two-sided) was conducted to compare the relationship between time and relative abundance and occurrence frequency of Rank I ARGs (relative abundance: $p < 1.8e\text{-}05$, occurrence frequency: $p < 0.00023$). The classification of Rank I ARGs was based on a comparison with the previously reported list (Supplementary Data 3)[26]. **g** Fast expectation-maximization for microbial source tracking (FEAST) estimating the source contribution of various habitats to the soil. Sample sizes were not normalized across habitats due to the limitations of sample numbers in some habitats. All habitat pairs were analyzed one-to-one, therefore the results indicate the proportion of shared ARGs. All habitat pairs were counted 999 times. **h** The presence and relative abundance of high-risk ARG subtypes detected in soil and their association with time. The relative abundance of Rank I ARGs was normalized to a range of 0 to 1 in each period. * indicates $p < 0.05$, $R^2$ is calculated from the linear model, and $r$ is calculated from the Pearson correlation (liner regression). The correlation was statistically tested using both Pearson's correlation test (two-sided) and a linear model (two-sided). *TEM-156*: $p < 0.041$, *erm(B)*: $p < 0.026$, *ANT(2")-Ia*: $p < 0.043$, *AAC(3)-VIa*: $p < 0.033$, *lsa(E)*: $p < 0.041$, *ANT(6)-Ia*: $p < 0.025$, *mph(B)*: $p < 0.02$, *tet(M)*: $p < 0.016$, *dfrB1*: $p < 0.034$. Period A: 2008–2010, Period B: 2011–2013, Period C: 2014–2016, Period D: 2017–2019, Period E: 2020–2021.

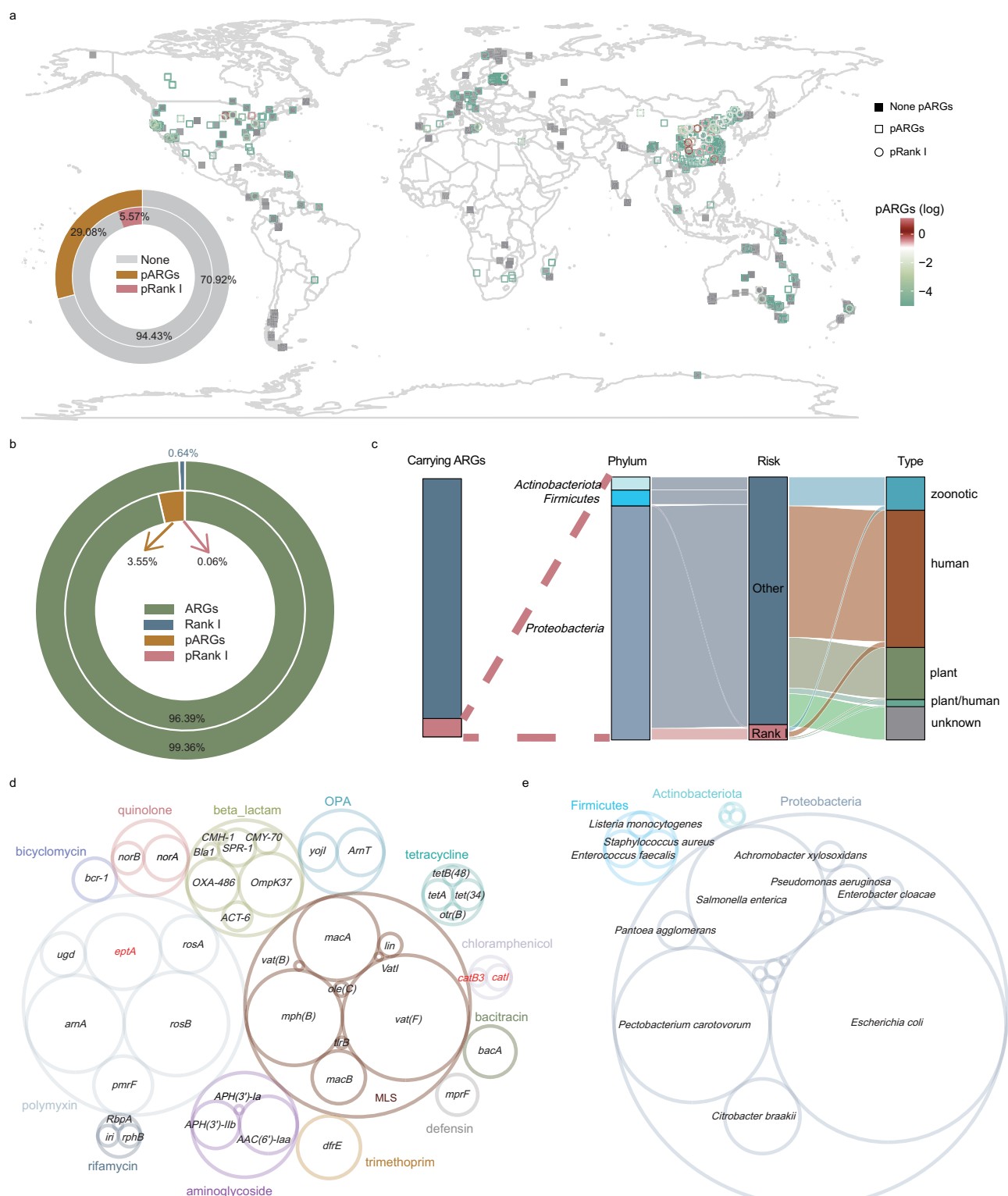

**Fig. 2 | The profile of pathogens carrying ARGs in soil. a** Distribution of prokaryotic pathogens carrying ARGs (pARGs) and Rank I ARGs (pRank I) across the global. ARGs carried by plasmids were excluded using geNomad[28], and only ARGs carried by chromosomes were included. Color represented the relative abundance of pARGs (gray: not detected, orange 0–10 copies per 1000 cells, red >10 copies per 1000 cells). The points for the islands are not marked on the map. The shape represents whether or not this pARGs and pRank I are detected (solid squares: the absence of pARGs and pRank, hollow squares: detection of pARGs only, hollow circle: detection of pRank I). The log transformation was performed on copies / 1000 cells. The outer circle of the pie chart represented the proportion of sample points for which pARGs were detected, and the inner circle represented the proportion of sample points for which pRank I were detected. **b** The proportion of pARGs and pRank I. The outer circle of the pie chart represented the proportion of Rank I ARGs to total ARGs at the contigs level. The inner circle of the pie chart represented the proportion of pARGs and pRank I to total ARGs at the contigs level. **c** Overall characteristics of prokaryotic pathogens in soil, including ARG carriers and non-carriers, taxonomy (phylum), level of risk, and pathogenicity. Only ARGs carried by chromosomes were included. **d** The profile of pARGs. Rank I ARGs were highlighted in red. **e** The profile of pathogens carrying ARGs on chromosomes.

samples, higher relative abundance of pARGs and pRank I were observed in the eastern coastal regions of North America and Asia. At the contig level, Rank I ARGs accounted for 0.64% of the total relative abundance of ARGs. A total of 3.55% of the ARGs were pARGs, of which 0.06% were pRank I (Fig. 2b). A total of 103 prokaryotic pathogens were detected, hosting 67 gene types of ARGs across 19 species. These species belonged to *Proteobacteria* (13 species), *Firmicutes* (3 species), and *Actinobacteriota* (3 species) (Fig. 2c). All of these pathogens carrying Rank I ARGs were capable of infecting humans, with *catI*, *catB3*, and *eptA* being the predominant pRank I types (Fig. 2d). Our results suggested that *E. coli* was the most abundant pathogen carrying ARGs on chromosomes, accounting for 46.0%, followed by a plant pathogen *Pectobacterium carotovorum* (27.2%) (Fig. 2e). Overall, pARGs, especially pRank I, had a high occurrence frequency in soil and *E. coli* was the predominant prokaryotic pathogen in soil.

## The profile for *E. coli* isolation genomes

As *E. coli* was one of the most important pathogens carrying ARGs in soil, it was selected as a representative of soil prokaryotic pathogens. We collected the genomes of 9700 *E. coli* strains isolated from soil, human, chicken feces, cattle feces, and swine feces (*E. coli* from specific habitats were referred to the habitat source *E. coli*, such as soil source *E. coli*) from the NCBI database. After filtering, 1312 genomes were excluded due to a lack of isolation information or misclassification (Fig. S10). A total of 8388 genomes, isolated between 1977 and 2023, were used for further study (Figs. S10 and S11 and Supplementary Data 8). Meanwhile, given their importance in soil (Fig. 1), we focused on Rank I ARGs and mobile Rank I ARGs (MRank I ARGs), which have mobile genetic elements (MGEs) located within 5 kb upstream and downstream of the Rank I ARGs[22]. The same database (SARG3.0_S) used for metagenomic ARGs analysis was conducted, and genes related to multidrug efflux pumps were also excluded. Firstly, we constructed the rarefaction curves for *E. coli* isolate genomes, which showed that the rarefaction curves reached the plateau for the studied habitats and clearly reflected the diversity of ARGs in these habitats (Fig. S12).

We focused on the temporal trends in the mean copy number of Rank I and MRank I ARGs, the occurrence frequency of MRank I ARGs (number of Rank I / MRank I gene types), and their richness in soil source *E. coli* (Fig. 3a, b). The results showed a significant increase in all parameters over time ($p < 0.001$, r = 0.69–0.85), indicating a growing potential ARGs risk associated with soil source *E. coli* each year. Even after rarefying based on the same sample size, the consistent trend remained (Fig. S13). To better understand ARGs risks across different habitats, we compared Rank I ARGs *E. coli* from various sources (Fig. 3c, d). We further performed period-based subsampling, aligned with metagenomic groupings, to ensure that each habitat and period had equal data volumes, eliminating the effect of sample size. Results confirmed that the mean copy numbers and richness of Rank I and MRank I ARGs for per soil *E. coli* genome showed a highly significant increasing trend over time, similar to that of swine and human *E. coli* (Fig. 3c, d). *E. coli* from chicken, on the other hand, showed an increasing and then decreasing trend, while those from cattle exhibited no correlation with time. Among all habitats, *E. coli* genomes from soil showed the greatest variation in both mean copy numbers and richness when comparing post-2020 with pre-2007. Mean copy numbers increased by 2.6 to 6.9 times, while richness rose by 1.9 to 5.5 times. A total of 30 Rank I ARGs (belonging to 30 subtypes and 9 types) were observed in soil *E. coli*, conferring resistance to the antibiotic classes of aminoglycoside, beta-lactam, tetracycline, and macrolide-lincosamide-streptogramin (MLS) (Fig. 3e). Among all Rank I ARGs and MRank I ARGs, the highest relative abundance were observed for *eptA* and *tet(B)*, which were detected in all periods. Overall, these results indicated that, in the typical pathogens, the risk of ARGs in

soil has also increased year by year, consistent with the metagenomic analysis.

## The association of soil Rank I ARGs with other habitats

Metagenomic analysis suggested that soil may be a potential sink for Rank I ARGs in humans and livestock (Fig. 1). To validate this hypothesis, FEAST analyses of the Rank I ARG composition in *E. coli* isolate genomes were performed to assess the influence of human and livestock (potential sources) on soil (a potential sink) across different periods (Fig. 4a, b)[27]. Briefly, human source (26.6%) and chicken source (25.3%) contributed the most to soil source, followed by cattle source (23.9%) (Fig. 4a). Remarkably, the relative importance of different sources showed significant variation across different periods (Fig. 4b). The contributions of *E. coli* from human and chicken sources to soil showed a significant upward trend, while those from cattle and swine sources exhibited a decline. Human contributions surpassed those from chicken in Period B (2011–2013) and have remained the dominant source for soil *E. coli* through the most recent period (Period E, post-2020) (Fig. 4b).

To further explore the association between Rank I ARGs carried by *E. coli* from different sources and soil source *E. coli*, we introduced an evaluation standard to assess their connectivity. Briefly, this method calculated the degree of habitat alternation for identical Rank I ARGs (gene level) on a phylogenetic tree and normalizes it against theoretical extremes to determine the connectivity between other habitats and soil (Fig. 4c). Since *eptA* is the dominant Rank I ARG gene conferring resistance to polymyxin in *E. coli*, we used this gene as an example to calculate the connectivity of Rank I ARGs between different habitats and soil. Results showed that the highest level of connectivity could be observed between human and soil (connectivity = 0.28), followed by chicken (connectivity = 0.25), and swine (connectivity = 0.21). Consistent with the FEAST results, we also observed a significant increase in the connectivity between humans and soil over time, exhibiting an almost linear relationship ($p < 0.001$, $R^2 = 0.91$). These results underscored that soil was an important node for the dissemination of Rank I ARGs. Moreover, *E. coli* resistome for human sources and soil sources were highly associated, both in terms of potential relationship (calculated by FEAST) and connectivity.

## Genome pairs with identical ARG sequences across different habitats

In light of the observed connectivity between soil and other habitats, we further examined the proportion of identical ARG sequences between *E. coli* genomes from soil and other habitats. We screened all 8388 genomes for ARGs genetic blocks (complete open reading frames, ORFs) of 100% identical sequences shared between any pairs of genomes from different habitats. To avoid inflating estimates of the number and frequency of sequence sharing events, we defined these events as the proportion of genome pairs that share at least one similar sequence, rather than using the absolute count of distinct BLAST sites. To ensure the reliability of the results, we randomly selected 100 genomes from each habitat for each calculation and repeated this process 999 times (Fig. 5a and Fig. S14). Only genome pairs that included soil sources *E. coli* were focused, and a total of 44,905,050 calculations were performed. Results showed that the proportion of sequence sharing events between soil and other habitats reached 8.7% for total ARGs and 0.8% for Rank I ARGs (Fig. 5b).

The similarity between sequences may result from both vertical gene transfer (VGT) and horizontal gene transfer (HGT). To distinguish them, we compared single-nucleotide polymorphisms (SNPs) in the vertically transmitted, slow-evolving genes of each genome pair[29]. The DNA sequences corresponding to the 120 ribosomal genes were extracted via GTDB-Tk, the maximum length was 175,318 bp and the average length was 161,396 bp (Fig. S15). A molecular clock of 1 SNP/genome/year and a genome size of $10^6$ bp was assumed[30,31], which is a

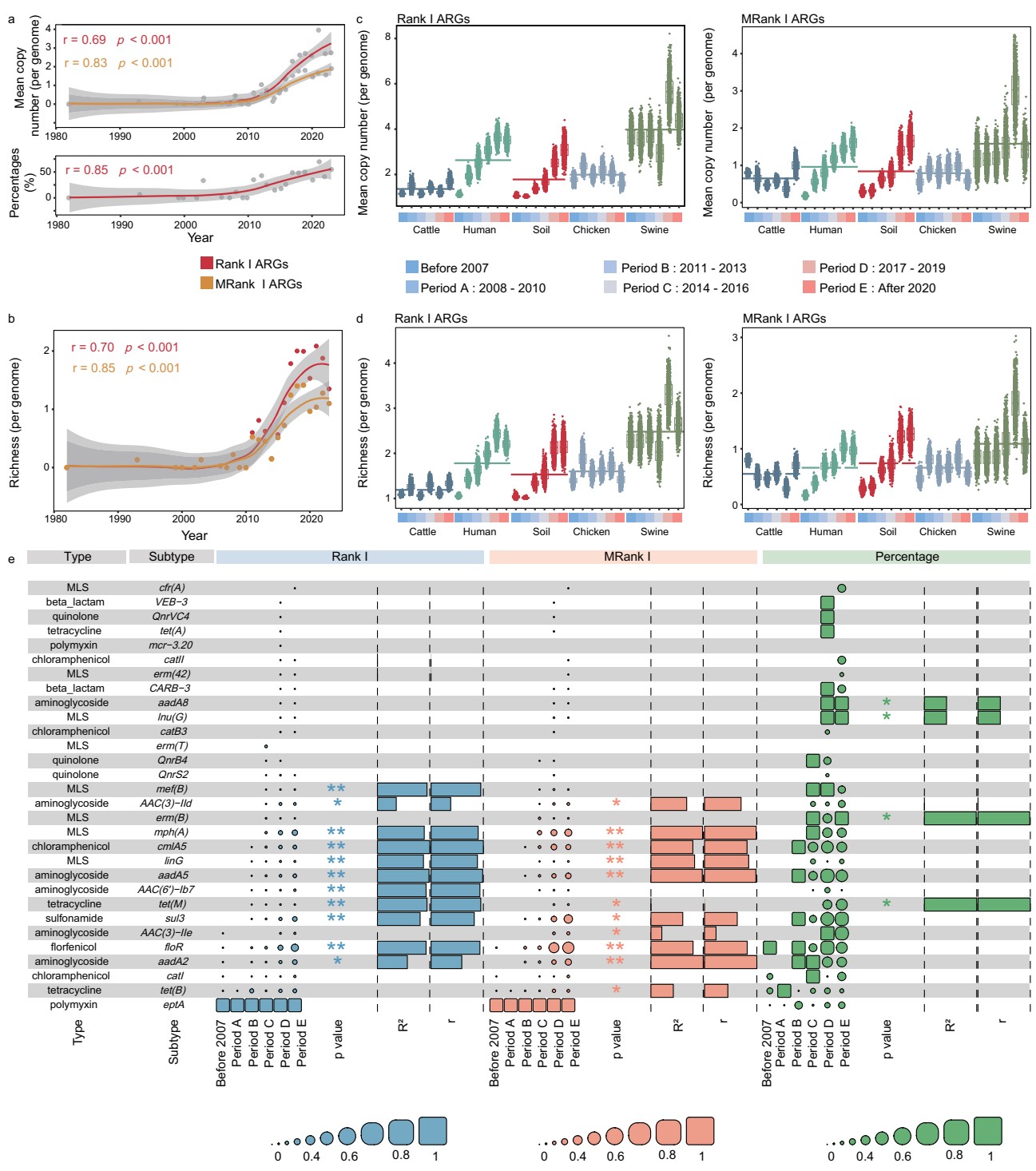

strict standard for *E. coli*[32]. The single-copy protein could generate up to 8.3 SNPs over a 47-year period (1977–2023, the years of genome collection) across 175,318 bp. We set a threshold of 1000 SNPs to distinguish whether the sequence sharing events were caused by VGT along bacterial lineages (SNPs < 1000) or HGT (SNPs ≥ 1000). This threshold, significantly higher than the 8 SNPs expected from VGT, ensures the conservativeness of the results. It showed that more than 40.7% of genome pairs exhibited sequence similarity that could be considered indicative of HGT, while the remaining 59.3% of genome pairs were classified as VGT (Fig. 5c). Additionally, when comparing SNP thresholds of 10 and 100, we found that the 10 SNP threshold filtered only 2.9% of bacterial pairs, and the 100 SNP threshold filtered 3.4%, further supporting the stringency of the 1000 SNP threshold (Fig. S16). Indeed,

nearly 95.0% of the Rank I ARGs involved in potential HGT events had MGEs located within 5 kb upstream and downstream (Fig. S17).

The proportion of sequence sharing events occurring in genome pairs with over 1000 SNPs was the highest between human-soil and chicken-soil, accounting for 53.3% and 51.2%, respectively (Fig. 5d). To visualize the phylogenetic distances between genomes from different habitats, four phylogenetic trees were constructed, including cattle-soil (Fig. 5e), human-soil (Fig. 5f), chicken-soil (Fig. 5g), and swine-soil (Fig. 5h) pairs. The results revealed distinct habitat-specific characteristics in the phylogenetic distribution of *E. coli* genomes from human and soil sources (Fig. 5f). Given the highest number of sequence sharing events between human and soil genome pairs, along with distinct habitat-specific features in the genomic phylogenies, we

**Fig. 3 | Rank I ARGs copy number and richness in the genomes of 8388 *E. coli* isolates. a** Mean of *E. coli* Rank I ARGs (per genome), mobile Rank I ARGs (MRank I ARGs), and the occurrence frequency of MRank I ARGs (genomes existing MRank I ARGs/ total genomes) over time in soil (1404 genomes). Each point represented the mean of all samples for that year. Gray shading denotes the 95% confidence intervals. Pearson's correlation (two-sided) with loess regression for data smoothing was used to test the statistical relationship between variables (mean copy number for Rank I ARGs: $p < 0.00018$, mean copy number for MRank I ARGs: $p < 6.7e\text{-}07$, occurrence frequency of MRank I ARGs: $p < 1.9e\text{-}07$). **b** Richness of *E. coli* Rank I ARGs (per genome) and MRank I ARGs over time in soil (1404 genomes). The richness was calculated in gene level. Each point represented the average of all samples for that year. Gray shading denotes the 95% confidence intervals. Pearson correlation (loess regression) was conducted to obtained the *p* value and *r*. Richness refers to the number of Rank I and MRank I ARGs gene type. Pearson's correlation (two-sided) with loess regression for data smoothing was used to test the statistical relationship between variables (diversity for Rank I ARGs: $p < 0.00015$, diversity for MRank I ARGs: $p < 1.3e\text{-}07$). **c** Mean copy numbers of Rank I ARGs and MRank I ARGs in different habitats across various periods (using 8,388 genomes collected from various periods). 999 rounds of random sampling based on the minimum habitat and period counts were conducted to eliminate the impact of sample size differences across different years on the results. 38 genomes per habitat per period were included in the analysis. Each point represented the average of all samples for that year. "Before 2007" represented genomes collected prior to 2007, and the remaining period groupings were consistent with that in metagenomic analysis (i.e., Period A: 2008–2010, Period B: 2011–2013, Period C: 2014–2016, Period D: 2017–2019, Period E: After 2020). In the boxplots of panels, hinges indicate the 25th, 50th, and 75th percentiles, whiskers indicate 1.5× interquartile ranges, dots indicate the average value from each random sampling.

Significant comparisons (two-sided *t* test) between different temperature and different groups were calculated and detailed *p* value and T stat were provided in the Supplementary Data 9. **d** Richness of Rank I ARGs and MRank I ARGs in different habitats across various periods (using 8388 genomes collected from various habitats). 999 rounds of random sampling were performed and 38 genomes per habitat per period were included. Significant comparisons (two-sided *t* test) between different temperature and different groups were calculated and detailed *p* value and T stat were provided in the Supplementary Data 10. In the boxplots of *p* panels, hinges indicate the 25th, 50th, and 75th percentiles, whiskers indicate 1.5× interquartile ranges, dots indicate the average value from each random sampling. Richness refers to the number of Rank I and MRank I ARGs gene type. **e** The Rank I ARGs copy number, MRank I ARGs copy number, and the occurrence frequency of MRank I ARGs (genomes existing MRank I ARGs/ total genomes) in soil collected from different period. The Rank I ARGs copy number, MRank I ARGs copy number, and the occurrence frequency of MRank I ARGs were normalized to a range of 0 to 1 in each period. $R^2$ is calculated from the linear model, and r is calculated from the Pearson correlation (liner regression). * indicated $p < 0.05$, ** indicated $p < 0.01$. The correlation was statistically tested using both Pearson's correlation test (two-sided) and a linear model (two-sided). Rank I ARGs: *catlI*: $p < 0.048$, *erm(42)*: $p < 0.048$, *mef(B)*: $p < 0.0074$, *AAC(3)-IId*: $p < 0.028$, *mph(A)*: $p < 0.0086$, *cmlA5*: $p < 0.0062$, *linG*: $p < 0.0090$, *aadA5*: $p < 0.0065$, *AAC(6')-Ib7*: $p < 0.0078$, *tet(M)*: $p < 0.0074$, *sul3*: $p < 0.010$, *floR*: $p < 0.0078$, *aadA2*: $p < 0.018$. MRank I ARGs: *AAC(3)-IId*: $p < 0.013$, *mph(A)*: $p < 0.0052$, *cmlA5*: $p < 0.0096$, *linG*: $p < 0.0087$, *aadA5*: $p < 0.0053$, *tet(M)*: $p < 0.047$, *sul3*: $p < 0.015$, *AAC(3)-IIe*: $p < 0.034$, *floR*: $p < 0.0095$, *aadA2*: $p < 0.0049$, *tet(B)*: $p < 0.021$. occurrence frequency of MRank I ARGs: *aadA8*: $p < 0.041$, *lnu(G)*: $p < 0.041$, *AAC(3)-IId*: $p < 0.045$, *erm(B)*: $p < 0.036$, *tet(M)*: $p < 0.036$.

suggested the possibility of cross-habitat horizontal gene transfer (HGT) between humans and soil. We further analyzed the temporal distribution patterns of sequence similarity events between various habitats and soil (Fig. 5g). The results showed a significant increasing trend in the proportion of identical ARGs between all habitats and soil, including bacterial pairs within the soil. Although the proportion of sequence similarity events in total ARGs between human-soil and chicken-soil was similar (11.7%), the proportion of Rank I ARGs between human-soil genome pairs (1.4%) was significantly higher than in other habitats (0.2%–0.6%). Detailed ARGs profile on the sequence sharing events of total ARGs, and Rank I ARGs for different habitats were shown in Fig. S18. Aminoglycoside, MLS, and beta-lactam were the most predominant types of Rank I ARGs in which HGT occurred between human and soil source *E. coli* (Fig. 5h). Overall, the sequence similarity of Rank I ARGs between humans and soil was significantly higher than that of other habitats and increased year by year. Cross-habitat HGT may be one of the important factors contributing to the entry of Rank I ARGs from various sources into the soil.

### The relationship of antibiotic resistome between soil and human clinical

To further confirm the relationship of antibiotic resistome between soil and human, we collected the datasets for human clinical antibiotic resistance from five public datasets (i.e., Resistancemap, European Centre for Disease Prevention and Control Surveillance Atlas, PLISA Health Information Platform for the Americas, World Health Organization, and China Antimicrobial Resistance Monitoring System) and separate datasets were constructed for all pathogenic bacteria and *E. coli* alone. The dataset for all pathogenic bacteria covered the period from 1998 to 2022 and included 126 countries, 18 pathogens, and 53 antibiotic agents, with the *E. coli* dataset being a subset of the total pathogenic bacteria. Based on the Pearson correlation and linear model, we assessed the relationship between the soil metagenomic data and human clinical antibiotic resistance. Over time, all detected data (i.e., relative abundance of Rank I ARGs in soil metagenomic samples, occurrence frequency of Rank I ARGs in soil metagenomic samples, Rank I ARGs copies in *E. coli* isolates, MRank I ARGs copies in *E. coli* isolates, HGT efficiency for total ARGs in

human-soil, and HGT efficiency for Rank I ARGs in human-soil) were highly correlated with the datasets for all pathogenic bacteria and *E. coli* ($r = 0.66$–$0.95$, $R^2 = 0.40$–$0.89$, $p < 0.001$) (Fig. 6a–f). Remarkably, no significance was observed between the relative abundance of total ARGs and human clinical antibiotic resistance (Fig. S19). We further mapped Rank I ARGs in soil and found that Rank I ARGs were ubiquitous, being detected in most samples from North America, Asia, Europe and Oceania (Fig. 6g). Overall, these results demonstrated a consistent pattern of change between soil resistance groups and human resistance from multiple aspects, and emphasized the importance and urgency of focusing on soil Rank I ARGs.

## Discussion

Based on 3965 metagenomic sequencing datasets and 8,388 *E. coli* isolate genomes, we found that both the relative abundance and mobility of Rank I ARGs have increased over time. Additionally, the soil antibiotic resistome became increasingly related to the human antibiotic resistome. Moreover, the robust and positive correlation between soil and clinical antibiotic resistance highlighted that soil played a crucial role in the One Health framework for comprehending global antibiotic resistome, especially those relevant to human health.

Although ARGs are almost ubiquitous in soil, only a few might pose a high risk to human health[33]. Rank I ARGs are representative of these high-risk ARGs because of their human association, mobility, and presence in pathogens[23]. Our results suggested an increasing risk of soil ARGs, as the relative abundance and occurrence frequency of Rank I ARGs in soil metagenomic samples, as well as the Rank I ARGs and MRank I gene copy numbers in *E. coli* isolate genomes have already increased by 2.0–36.3 times than pre-2010 levels (Figs. 1 and 3). Archived soil analysis has also demonstrated a growing abundance of ARGs since 1940[34]. Indeed, this growing trend might be attributed to the advancement of sequencing technologies and the number of samples[35]. However, the rarefaction for both soil metagenomic sequencing data and *E. coli* isolate genomes plateaued (Figs. S3 and S12), suggesting that the selected samples were representative of the soil ARG resistomes[36]. Continuous introduction of bacteria carrying

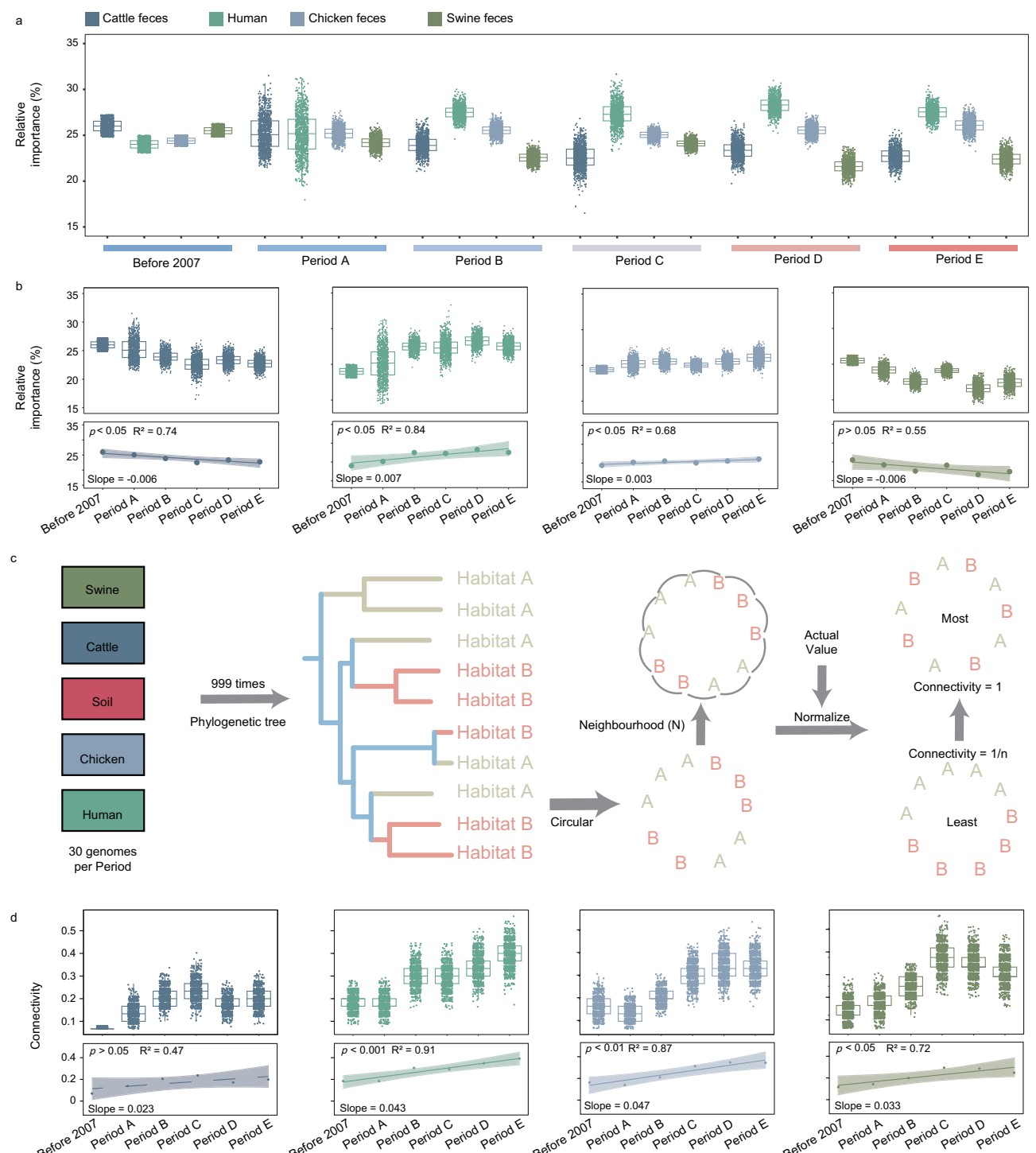

Rank I ARGs, HGT driven by environmental stress, trace antibiotic residues, and co-selection caused by various pollutants might be the potential mechanisms for the existence of Rank I ARGs in soil.

The soil ARG antibiotic resistome was significantly different from other habitats (Supplementary Data 4 and 5), likely attributable to various abiotic factors[37]. Considering that high antibiotic pressures are indeed uncommon in soil, even in agricultural soils[38], it is widely recognized that the soil antibiotic resistome is influenced by environmental factors (e.g., pH[14], and total organic carbon[39]), location (i.e., longitude and latitude)[12], vegetation[40], and climatic seasonality (i.e., cold weather)[11]. However, Rank I ARGs might be an exception. Rank I ARGs could be a key parameter for characterizing the relationship between human-related antibiotic resistome and the soil antibiotic

resistome, as genes from non-human related environments were excluded from the selection basis of Rank I ARGs[23]. Meanwhile, our results provided a body of evidence that (i) no unique Rank I ARGs (biomarker Rank I ARGs gene) were present in soil (Supplementary Data 6), and (ii) both the metagenomic data and *E. coli* isolate genomes confirmed that humans and livestock were the main sources of Rank I ARGs in soil (Figs. 1 and 4).

According to the results obtained by metagenomic sequencing and pure culture, we found that (I) the composition and trend of Rank I ARGs in the resistome of humans and soil showed similar patterns (Figs. 1 and 3), (II) the highest connectivity could be observed between the Rank I ARGs hosted by soil and human source *E. coli* (Fig. 4), (III) the highest sequence similarity events could be observed between

**Fig. 4 | Attribution of soil Rank I ARGs and the connectivity of soil with other habitats. a** The attribution of soil Rank I ARGs. Total 8388 *E. coli* isolates were included in the FEAST analysis. 999 rounds of random sampling were carried out, and 38 genomes per habitat per period were included in each calculation. In the boxplots of panels, hinges indicate the 25th, 50th, and 75th percentiles, whiskers indicate 1.5× interquartile ranges, dots indicate the average value from each random sampling. Significant comparisons (two-sided *t* test) between different temperature and different groups were calculated and detailed *p* value and T stat were provided in the Supplementary Data 11. **b** The impact of different habitats on the composition of soil Rank I ARGs. In the boxplots of panels, hinges indicate the 25th, 50th, and 75th percentiles, whiskers indicate 1.5× interquartile ranges, and dots indicate the average value from each random sampling. Nine hundred and ninety-nine rounds of random sampling were carried out, and 38 genomes per habitat per period were included in each calculation. Significant comparisons (two-sided *t* test) between different temperature and different groups were calculated

and detailed *p* value and T stat were provided in the Supplementary Data 11. The correlation was statistically tested using linear model (two-sided). $R^2$ and slope is calculated from the linear model. Cattle feces: *p* < 0.017, human: *p* < 0.026, chicken feces: *p* < 0.034. **c** Workflow for calculating connectivity. **d** The connectivity of soil with other habitats over time. 999 rounds of random sampling were carried out, and 30 genomes per habitat per period were included in each calculation. The results used *eptA* (one gene type of Rank I ARGs) as an example. In the boxplots of panels, hinges indicate the 25th, 50th, and 75th percentiles, whiskers indicate 1.5× interquartile ranges, dots indicate the average value from each random sampling. Significant comparisons (two-sided *t* test) between different temperature and different groups were calculated and detailed *p* value and T stat were provided in the Supplementary Data 12. The correlation was statistically tested using linear model (two-sided). $R^2$ and slope is calculated from the linear model. Human: *p* < 0.0034, chicken feces: *p* < 0.0065, pig feces: *p* < 0.031.

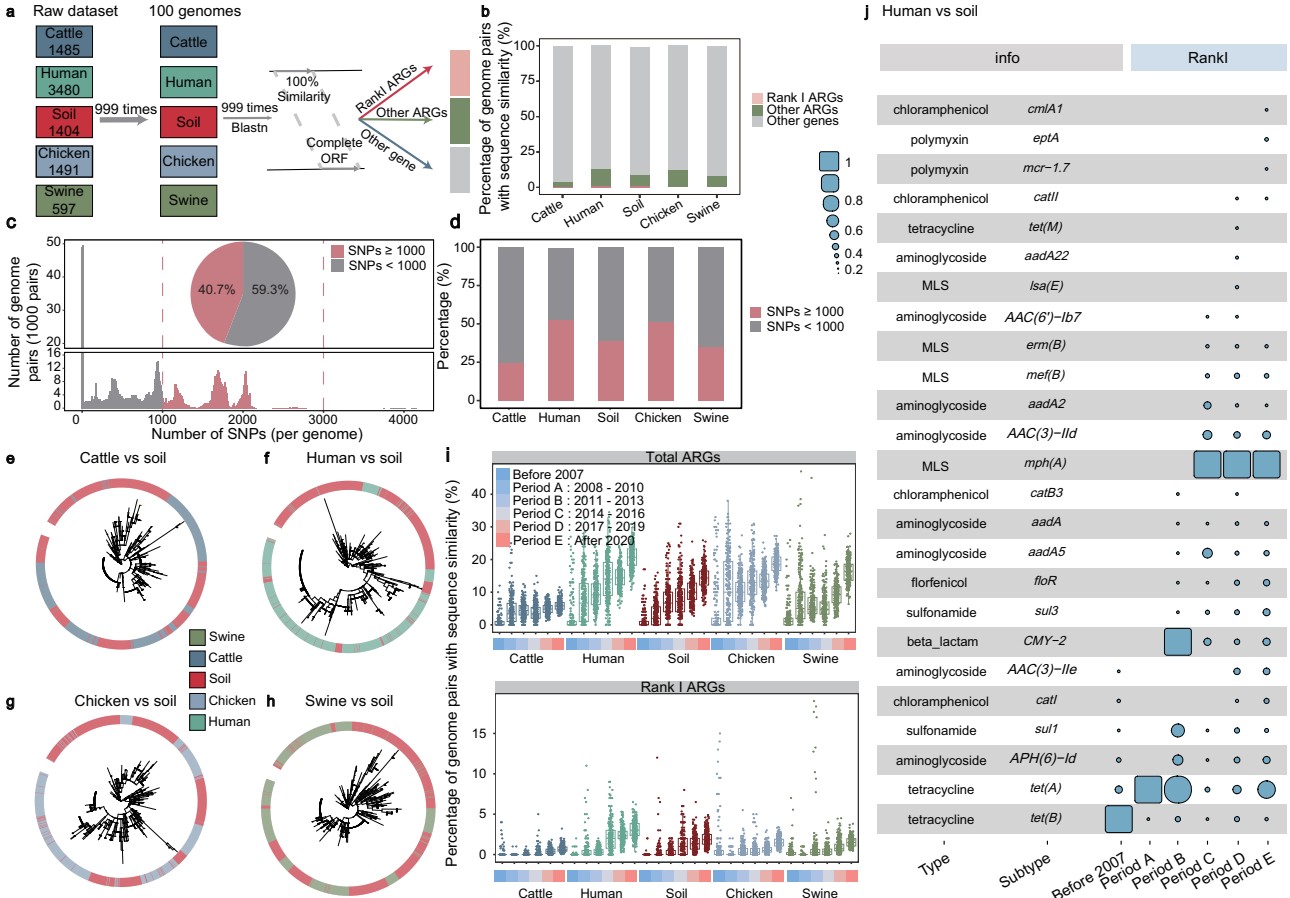

**Fig. 5 | Genome pairs of identical ARGs in soil and other habitats. a** Workflow for finding genome pairs with identical ARGs. **b** Proportion of genome pair with identical ARGs between soil-soil and soil-other habitats. Soil indicated that the genome pairs were both from the soil habitat. **c** SNPs between various genome pairs. The pie chart represented the proportion of SNPs with counts ≥1000 and <1000. All calculations (44,905,050 pairs) were included. **d** Percentage of VGT (SNP < 1000) and HGT (SNP ≥ 1000) in various habitats. **e** Phylogenetic tree of genomes isolated from soil and cattle with SNP ≥ 1000. **f** Phylogenetic tree of genomes isolated from soil and humans with SNP ≥ 1000. **g** Phylogenetic tree of genomes isolated from soil and chicken with SNP ≥ 1000. **h** Phylogenetic tree of genomes isolated from soil and

swine with SNP ≥ 1000. **i** Proportion of genome pair with identical ARGs between soil-soil and soil-other habitats in different periods. Soil indicated the genome pairs were both from the soil habitat. In the boxplots of panels, hinges indicate the 25th, 50th, and 75th percentiles, whiskers indicate 1.5× interquartile ranges, and dots indicate the average value from each random sampling (*n* = 999 rounds of random sampling). Significant comparisons (two-sided *t* test) between different habitats and different periods were calculated, and detailed *p* value and T stat were provided in the Supplementary Data 13. **j** Changes in identical Rank I ARGs copy numbers between soil and human across different periods. The copy numbers of Rank I ARGs was normalized to a range of 0 to 1 in each period.

genome pairs isolated from soil and human (Fig. 5). Meanwhile, all of these patterns showed significant increases over time. Thus, all evidence suggested that the soil antibiotic resistome was highly associated with human antibiotic resistome, and this association increased over time. Undeniably, livestock also had a significant impact on soil.

Before Period B (2011–2013), the resistome of chickens and cattle had the greatest influence on soil, but this influence was gradually replaced by that of humans till now. This might be due to heightened human activity, increased human antibiotic use[41], and stricter antimicrobial regulations in livestock rearing[42]. Such as banning the use of

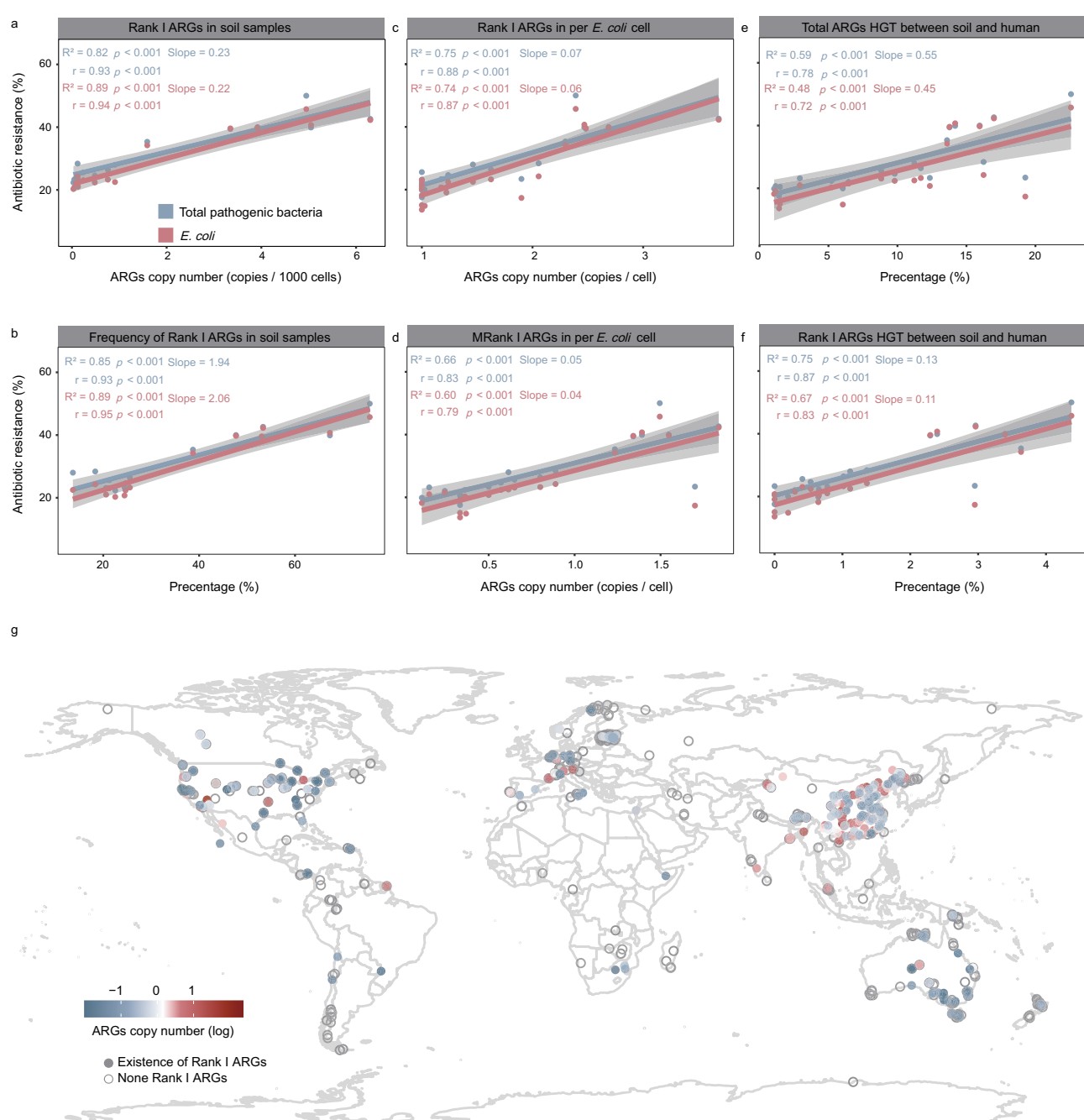

**Fig. 6 | The relationship between soil antibiotic resistome and human clinical antibiotic resistance. a** The relationship between the relative abundance of Rank I ARGs in soil metagenomic data and human clinical antibiotic resistance. Gray shading denotes the 95% confidence intervals. The correlation was statistically tested using both Pearson's correlation test (two-sided) and a linear model (two-sided). Total pathogenic bacteria: $p < 2e-06$, *E. coli*: $p < 2e-06$. **b** The relationship between the occurrence frequency of Rank I ARGs in soil metagenomic data (the number of samples with Rank I ARGs detected in each year / total sample number in that year) and human clinical antibiotic resistance. Gray shading denotes the 95% confidence intervals. The correlation was statistically tested using both Pearson's correlation test (two-sided) and a linear model (two-sided). Total pathogenic bacteria: $p < 2e-07$, *E. coli*: $p < 2.5e-07$. **c** The relationship between the copy number of Rank I ARGs in *E. coli* genome and human clinical antibiotic resistance. Gray shading denotes the 95% confidence intervals. The correlation was statistically tested using both Pearson's correlation test (two-sided) and a linear model (two-sided). Total pathogenic bacteria: $p < 2.1e-07$, *E. coli*: $p < 4e-06$. **d** The relationship between the

copy number of MRank I ARGs in *E. coli* genome and human clinical antibiotic resistance. Gray shading denotes the 95% confidence intervals. The correlation was statistically tested using both Pearson's correlation test (two-sided) and a linear model (two-sided). Total pathogenic bacteria: $p < 3.2e-07$, *E. coli*: $p < 2.3e-05$. **e** The relationship between the proportion of total ARGs displaying HGT and human clinical antibiotic resistance. Gray shading denotes the 95% confidence intervals. The correlation was statistically tested using both Pearson's correlation test (two-sided) and a linear model (two-sided). Total pathogenic bacteria: $p < 2.4e-05$, *E. coli*: $p < 0.00026$. **f** The relationship between the proportion of Rank I ARGs displaying HGT and human clinical antibiotic resistance. Gray shading denotes the 95% confidence intervals. The correlation was statistically tested using both Pearson's correlation test (two-sided) and a linear model (two-sided). Total pathogenic bacteria: $p < 2.4e-07$, *E. coli*: $p < 3.1e-06$. **g** The global map of the relative abundance of Rank I ARGs in soil based on the metagenomic data (short reads). White is the area where no data was collected. The points for the islands are not marked on the map. The log transformation was performed on copies / 1000 cells.

antimicrobial growth promoters in the European Union in 2006 and the People's Republic of China banning the use of fluoroquinolones (http://www.moa.gov.cn/nybgb/2015/jiuqi/201712/t20171219_6103873.htm) and colistin (http://www.moa.gov.cn/nybgb/2016/dibaqi/201712/t20171219_6102822.htm) in animals in 2016 and 2017, respectively.

The antibiotic resistomes of different habitats have always been linked to microbial population phylogenies along ecological gradients. However, several clinically important resistance genes could cross habitat boundaries[43,44]. Based on potential attribution analysis, feces were identified as the great reservoirs of Rank I ARGs (Fig. 1g). In general, human and animal waste ultimately enter the sewer system, leading to water contamination. Yin et al. suggested that rivers with high human density, advanced economic development, and intensive industrial activities are more vulnerable to human activities, such as sewage plant discharge, than those in less developed regions (by more than five times). Thus, these results implied that soil might act as an intermediary, the water as a disseminator[36]. The cross-habitat transmission pathways may involve both VGT along bacterial lineages and HGT. VGT is the more common way and can occur at interfaces where soil and humans overlap, such as farmlands[38], urban areas[45], and greenspaces[46]. In these confluence areas, environmental bacteria mix with bacteria from humans. Additionally, soil may also act as a medium for transfer between livestock and humans via the food chains, via direct contacts, or other environmental pathways[47]. This might explain the high connectivity and sequence similarity events observed between human and soil Rank I ARGs. However, a large number (40.7%) of *E. coli* isolate genome pairs (between soil and other habitats) with sequence similarity ARGs events had more than 1000 SNPs, suggesting the potential importance of HGT across habitats (Fig. 5). The SNPs between genome pairs were based on the DNA sequences corresponding to the 120 ribosomal genes, which are slow-evolving genes, and over 47 years (the time span of genome collection), they generated a maximum of 8.2 SNPs. An excessive number of SNPs suggested that these genome pairs with identical ARG sequences were more likely the result of HGT events rather than VGT along bacterial lineages. Under the assumed molecular clock, for 1000 bp HGT events (the average sequence lengthen in the SARGs database) with 100% similarity, it would take 1 year to accumulate $10^{-3}$ SNPs[29], meaning it would take 1000 years to accumulate 1 SNP and be filtered out from our analysis. Our 1000 bp ARGs HGT events without any mutations correspond to HGT events that occurred within the last 1000 years. Thus, VGTs and cross-habitat HGTs were the key factors enabling these Rank I ARGs to spread across habitat boundaries (i.e., from human to soil). The cross-habitat HGT cannot be attributed exclusively to transfers in confluence areas. Generalist taxa, which could persist or thrive in multiple habitats[48], may play a critical role in transferring ARGs between habitats, including *Aeromonas* spp. in WWTPs[49] and *Acinetobacter Johnsonii* in bioaerosol[50]. ARGs could jump to soil through these generalist taxa via a series of transmission events. The complex microbial community and environment of the soil also shape the conditions for HGT, including compound contamination, non-prokaryotic community members (e.g., protozoa and viruses)[51], and environmental stress. Indeed, enhanced connectivity among various environments and hosts across different geographical scales is another factor that facilitates the dissemination of ARGs[52]. VGT along bacterial lineages is a major factor in the spread of ARGs, while cross-habitat HGT may be an overlooked force driving the widespread occurrence of antibiotic resistance[29].

In addition to genomic evidence, we confirmed the significant associations between the soil antibiotic resistome and clinical human antibiotic resistance by constructing the global AMR datasets (Fig. 6). Our study provided solid evidence for the importance and risk of soil antibiotic resistance under the One Health framework. Due to limitations in the number of metagenomic samples, with uneven sample sizes across different regions within the same year, we could only explore their relationship over time. All the results collected from metagenomic samples and *E. coli* isolate genomes, including the relative abundance, number, and HGT of Rank I ARGs, showed a significant positive correlation between soil and the human antibiotic resistome, via linear model and the Pearson approach. Thus, our results were reliable in the temporal dimension, but they should also be confirmed on a global scale over time with additional data.

Although genes that cause resistance in clinical and veterinary settings are of greatest public health concern[53], environmental bacteria are also a reservoir of resistance with the potential to be transmitted to clinical pathogens[44,54]. The deliberate consumption of soil is common in farm animals, such as sheep[55] and dairy cows[56], and shapes their gut microbiome. Humans are also directly affected by the soil microbiome. For example, the skin and oral microbiome of farm workers are affected by the soil microbiome composition of the farms where they work[57]. Additionally, soil types correlate with the nasal and oral microbiome of the local population[58]. Due to the lack of detailed vegetation, it was unable to fully consider vegetation factors. Future research could focus on and document the vegetation status at sampling sites to deeply explore the impact of vegetation on soil ARGs resistome. Although soil is currently a sink for human ARGs, given the nature of soil and the many ways in which humans interact with it, we need to prevent and mitigate the potential for soil to become a source of ARGs in humans. Our analysis provided evidence of the relationship between the soil antibiotic resistome and human clinical antibiotic resistance within the One Health framework. These findings were derived from global data, offering broader applicability compared to smaller-scale or short-term studies.

## Methods

### Public metagenomic data collection

Briefly, a total of 3965 metagenomic samples from 12 habitats were used in this analysis. The datasets used in this study were from three sources: (1) 149 newly generated sequencing datasets, comprising 36 Chinese soil samples collected in 2018 and 113 samples collected in 2021 (Supplementary Data 1); (2) 2513 publicly available datasets which were reanalyzed in this study, including 2391 soil metagenomic datasets (Supplementary Data 1) and 122 crop surface metagenomic datasets (Supplementary Data 2); (3) 1303 publicly available datasets with previously reported results[36], spanning various habitats such as swine feces, cattle feces, chicken feces, human feces, water from wastewater treatment plants, sewage, natural sediment, natural water, marine water, and landfill (Supplementary Data 2). Overall, 2540 soil metagenomic samples were included in the analysis, of which 2391 samples were public data and 149 were in-house data (Supplementary Data 1). Land use and soil type for each sample were collected via ArcGis (v10.8) and were shown in Supplementary Data 1. To minimize possible bias, we only included public data that fulfilled the following criteria (i) Illumina shotgun data; (ii) paired-end data with FASTQ format; (iii) over 1 Giga Byte (GB); (iv) no culturing or any other additional experiments before DNA extraction; (v) not collected from potentially contaminated environments, including heavy metal contamination, coking sites, industrial waste, etc.; (vi) included detailed sample information (e.g., accurate coordinate and sampling time); (vii) average read length over 100 base pairs. These criteria represented an initial quality control of the soil dataset to minimize the influence of various sources on the results. The public data were downloaded using Aspera (https://www.ibm.com/cn-zh/products/aspera).

### In-house metagenomic data procedure

A total of 149 Chinese soil samples were collected in 2018 (36 samples) and 2021 (113 samples). Briefly, the upper soil layer was collected using a soil auger, and each sample was composed of five sub-samples (i.e., 4

corners and the center) to eliminate heterogeneity[59]. After removing plant roots and stones from the soil, the fresh soil was stored at −20 °C. DNA of these soil samples was extracted using DNeasy PowerSoil Kit (12888-50, Qiagen, Germany), following the manufacturers' instructions. The sequencing libraries were constructed following the product instructions of NEB Next® Ultra™ DNA Library Prep Kit for Illumina® (New England Biolabs, MA, USA) and sequenced by Illumina NovaSeq 6000 platform with a 150 bp paired-end reads strategy by a sequencing server (MAGIGENE Biological Technology Co. Ltd Guangzhou, China)[60]. Each sample yielded an average of 10 GB of raw data.

## Data analysis for metagenomic data

Both public metagenomic data (12 habitats) and in-house metagenomic data were analyzed using the same framework. In detail, Trimmomatic (v.0.36) was used to filter out low-quality reads (leading: 3, trailing: 3, slidingwindow: 4:20, minlen: 100)[61]. ARG profiles were quantified using ARGs-OAP (v3.2.2) with a cut-off of 80% identity, 75% coverage, and 1e-7 e-value[26]. The SARG3.0_S database was excluded sequences associated with transcriptional regulators (including activators and repressors), point mutations, and others. SARG3.0_S is provided as reference databases in the ARGs-OAP for the full analysis of environmental metagenomic datasets using similarity search algorithms[26]. In addition, considering the controversy surrounding multidrug efflux pumps, all genes related to these pumps were excluded to avoid possible mis-annotations of ARGs. The identified ARGs were classified at three levels (i.e., type, subtype, and gene levels (also known as gene types or reference sequences)), and the unit of copies of ARG per cell was used to normalize ARGs[62]. ARGs per cell (also equivalent to copy per genome if we assume one genome per cell) is the consensus unit[62], which normalizes sequencing depths, ARG lengths, and prokaryotic cells/genomes in the dataset since it is more suited to present the ARG relative abundance in samples. To classify Rank I ARGs, we compared our data against a previously reported list (Supplementary Data 3)[26]. We also tested and excluded the effect of fungi on the ARGs-OAP analyses (Text S5 and Table S3). The data was divided into 5 periods, including Period A: 2008–2010, Period B: 2011–2013, Period C: 2014–2016, Period D: 2017–2019, Period E: 2020–2021. To verify the reliability of the trends in total ARGs and Rank I ARGs, we performed subsampling 999 times for each period based on the smallest period sample size (Period A: 133), ensuring that all periods were evaluated with the same data volume. To further evaluate the influence of uneven country distribution across different periods, we fixed the number of continents included in each sampling (50 samples per time: 15 from Europe, 15 from America, 10 from Asia, and 10 from others) and performed 999 rounds of sampling (Fig. S9c). We also fixed the number of land use types included in each sampling (120 samples per time: 70 from farmland, 20 from forest, 10 from grassland, and 20 from others) and performed 999 rounds of sampling (Fig. S9d).

The clean data for 2540 soil metagenomic datasets (i.e., 149 in-house data and 2391 public data) were assembled by MEGAHIT (v1.2.9, parameters: k-min 35, k-max 115, k-step 20) to obtain contigs[63]. After filtering contigs less than 5000 bp, Prodigal (v2.6.3) was used to predict the ORFs of contigs. To identify the ARG-carrying contigs, the ORFs were aligned to the SARG3.0_S[26] via diamond blastp (v2.1.8.162)[64]. Before the annotation of the taxonomy, ARGs carried by plasmids were excluded using geNomad[28]. The ARG-carrying contigs were further annotated by Kraken2. According to the prokaryotic pathogens lists suggested by PHI-base (www.phi-base.org) and Catalog of human transmitted pathogenic microorganisms 2023, we focused on the ARG-carrying prokaryotic pathogens. Bowtie2 (version 2.4.2) was used for sequence mapping (very-sensitive), and Samtools (version 1.11) was used to process and convert the alignment results. After alignment, we calculated the copy/cell for pARGs and pRank I according to the formulas (cell number was obtained by ARGs-OAP).

$$\frac{copies}{cell} = \frac{1}{cell\ number} * \sum \frac{Reads\ counts * mapped\ length}{gene\ length}$$

## Public *E. coli* isolate genome collection

As *E. coli* is one of the most important prokaryotic pathogens in soil, we downloaded the genomes of 9700 *E. coli* isolates from NCBI. We only included the genomes of *E. coli* isolates that fulfilled the following criteria: (i) isolated from soil, human, chicken feces, cattle feces, and swine feces; (ii) clear isolation information, including isolation country and isolation date; (iii) similar genome number of different isolation sources. The taxonomy and phylogeny were further checked using the Genome Taxonomy Database Toolkit (GTDB-Tk; v2.4.0), resulting in the exclusion of 231 genomes. In total, 8388 *E. coli* isolate genomes were used, including 1404 from soil, 1485 from cattle feces, 1491 from chicken feces, 3411 from humans, and 597 from swine feces (Supplementary Data 8). These *E. coli* were isolated between 1977 and 2023 from 53 countries. The completeness of all genomes was >97%, and the contamination level was <5% (Supplementary Data 8).

## Data analysis for genomes of *E. coli* isolates

Prodigal (v2.6.3) was used to predict the ORFs of each *E. coli* genome. To obtain the profile of ARGs and MGEs, the ORFs of these genomes were aligned to the SARG3.0_S[26] and Mobile Genetic Element Database via diamond blastp (v2.1.8.162)[64], with the same cutoffs as described in "Data analysis for metagenomic data". The genes related to multidrug efflux pumps were also excluded. Rank I ARGs with mobile genetic elements (MGEs) present within 5 kb upstream and downstream are considered MRank I ARGs. Only complete ORFs were included in the following analysis. The occurrence frequency of MRank I ARGs is defined as the ratio of the number of genomes containing MRank I ARGs to the total number of genomes in that year (period). To verify the reliability of the trends in Rank I ARGs and MRank I ARGs, we performed subsampling 999 times for each period based on the smallest period sample size (period A of swine: 38 genomes), ensuring that all periods were evaluated with the same number of datasets (Fig. S15). Total 8388 *E. coli* isolates were included in the FEAST analysis in pure culture. 999 rounds of random sampling were carried out, and 38 genomes per habitat per period were included in each calculation.

## The calculation of connectivity

To further reveal the association between different sources *E. coli* and soil sources *E. coli*, connectivity was proposed (Fig. 4c). Connectivity compared the degree of habitat alternation for identical ARGs at the gene level on the phylogenetic tree. The selected ARGs needed to meet the following criteria: (I) existing in all periods and (II) being Rank I ARGs. Therefore, *eptA* was selected, as this gene is the most prevalent Rank I ARGs carried by *E. coli*. After extracting all complete sequences of *eptA*, we paired the sequences from different habitats and soil, randomly selecting 30 sequences from each habitat and period for 999 times. The sequences were aligned using ClustalW (v2.1), and phylogenetic trees were constructed with FastTree (v2.1). Connectivity was calculated based on the degree of habitat alternation between adjacent positions and normalized against theoretical extremes. The theoretical minimum value represents distinct ecological distribution between habitats (actual connectivity = 2), while the theoretical maximum represents complete alternation between habitats (actual connectivity = 60).

## Detection of sequence sharing events between genome pairs

Blastn (v2.6.0) was further used to screen blocks of DNA that were shared by two collected *E. coli* isolate genomes. Only DNA sequences with greater than 100% identity, and those that translate into complete ORFs, were retained. To prevent inflating estimates of sequence sharing counts and frequencies, we conservatively defined the sequence sharing events as the count of genome pairs sharing at least one identical ARGs[65], instead of focusing on the total number of unique blast hits between two genomes[29]. To eliminate the influence of different sample sizes in various habitats, we normalized the sample numbers in each habitat by randomly sampling 100 genomes in each habitat and repeated the analyses 999 times. About $4.5 \times 10^7$ genome pairs (only including soil and other habitats) were involved in the analysis. The detection of ARGs was the same as that described in "Data analysis for genomes of *E. coli* isolates" (Methods). We also extracted 5 kb regions flanking the ARGs involved in the sequence sharing events for annotating the associated MGEs (the same as "Data analysis for genomes of *E. coli* isolates" (Methods)).

## Detection of SNPs between genome pairs

We analyzed SNPs distribution in genes transmitted vertically and those evolving slowly to distinguish between HGT and vertical inheritance in the above sequence-sharing events (Detection of sequence sharing events between genome pairs). We identified and extracted the DNA sequences corresponding to the 120 ribosomal genes present in the genome by GTDB (Release 220) and GTDB-Tk (v 2.4.0). For each species pair with identified HGT candidates (totaling $4.5 \times 10^7$ genome pairs), we used snippy (v4.6.0) to extract SNPs and normalized the count to SNPs per $10^6$ base pairs. With a molecular clock of 1 SNP per $10^6$ bp per year[30,31,66], a genome fragment of 10,000 bp can reach a maximum sequence divergence of 1 SNP in 100 years[29]. For the sum of 120 ribosomal genes (175,319 bp), it will accumulate 18 SNPs in 100 years and 8.2 SNPs in 47 years (1977–2023). To ensure the rigor of the results, we set the threshold at 1000 SNPs, which is much higher than the 8.2 SNPs, although this may underestimate the HGT events. Therefore, when the number of SNPs in ribosomal genes between genome pairs exceeds 1000 SNPs, the ARGs involved in the sequence sharing events are considered to result from HGT. We also tested 100 SNPs and 10 SNPs as cutoffs, and results are shown in Fig. S18 for a reference.

## Datasets for human clinical antibiotic resistance

For this global analysis, we collected the human clinical antibiotic resistance genes from five sources, including Resistancemap (https://resistancemap.onehealthtrust.org/), European Centre for Disease Prevention and Control Surveillance Atlas (https://www.ecdc.europa.eu/en/surveillance-atlas-infectious-diseases), PLISA Health Information Platform for the Americas (https://opendata.paho.org/en), World Health Organization (https://dev-cms.who.int/initiatives/glass), and China Antimicrobial Resistance Monitoring System (https://www.carss.cn/sys/Htmls/dist/index.html). The dataset included 126 countries, 18 pathogens, 53 antibiotic agents, and was collected from 1998–2022. The definition of antibiotic resistance was that the tested strains were non-susceptible (i.e., intermediate or resistant) to an antibiotic. To eliminate the effect of sample size, isolation rates (antibiotic-resistant bacteria/total tested bacteria) were focused on. Only bacteria with a total number greater than 30 were included in the analysis.

## Reporting summary

Further information on research design is available in the Nature Portfolio Reporting Summary linked to this article.

## Data availability

The in-house metagenomic sequencing data generated in this study have been deposited in the National Center for Biotechnology Information (NCBI) Sequence Read Archive (SRA) database under accession number PRJNA1202346 and PRJNA1229199. All online metagenomic sequencing data and *E. coli* isolate genomes used in this study are available in the NCBI RefSeq database, IMG/M portal, and European Nucleotide Archive (Supplementary Data 1, 2, and 8). Source data are provided as a Source Data file. Source data are provided with this paper.

## Code availability

The script for the Connectivity Framework, and R script for heatmap and PCoA is publicly available on GitHub at https://github.com/Yuxiang-Zhao/ARGs or under Zenodo at https://doi.org/10.5281/zenodo.15826297.

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

## Acknowledgements

This work is supported by the National Natural Science Foundation of China (22193062 to T. Z., 22193061 to L. Z.). We thank the University of Hong Kong for the postdoctoral research fellowship and HKU Computer Center for providing the High Performance Computing System (HPC). We thank Kangfeng Li for his contribution in conceptualization. We thank James Voordeckers for his contribution to language editing. We thank Miaolian Hua for her contribution in sampling and data analysis. We thank Xinliang Xu for his contribution in providing the base map.

## Author contributions

Y.Z.: Conceptualization, Methodology, Visualization, Writing—Original Draft; L.L.: Writing—Review and Editing; Z. L., Y. H., X.Q., and S. L.: Methodology, Investigation; B. H. and L. Z.: Writing—Review and Editing; T. Z.: Writing—Original Draft, Funding acquisition, Conceptualization, Writing—Review and Editing.

## Competing interests

The authors declare no competing interests.
