## [Transparent Peer Review file · Nature Communications]

Global soil antibiotic resistance genes are associated with increasing risk and connectivity to human resistome

Corresponding Author: Professor Tong Zhang

Version 0:

Reviewer comments:

Reviewer #1

(Remarks to the Author)
Analysis

In the paper entitled “Deciphering the increasing risk of soil resistome associated with human resistance under One Health framework”, the authors perform an extensive analysis of antimicrobial resistance genes in more than 3,000 metagenome samples and use approximately 8,000 *E. coli* genomes from different environments to link the environment, animals, and humans. The authors use a rank metric to classify the resistance gene as high risk (Rank 1), intermediate, or low. This metric was developed and published by the group in 2021 in the paper doi:10.1038/s41467-021-25096-3. As their main conclusion, the authors state that soil is the main reservoir of Rank 1 ARGs that reach humans. This conclusion is based on a metric developed by the authors that they called “connectedness”. All this information, previous and new, made reading, understanding, and interpreting this paper challenging. Below, I leave my main comments regarding the manuscript.

As I anticipated, reading, understanding, and interpreting this study was challenging. There are many concepts that need to be understood in order to correctly interpret the results presented in this manuscript. The description of the public metagenome samples is referenced in paper no. 35 (Yin, X. et al. Global environmental resistome: Distinction and connectivity across diverse habitats benchmarked by metagenomic analyses. *Water Res.* 235, 2023.), but the sample numbers are different, which made it necessary to look at the supplementary material of this other paper. What is the sampling relationship between this study and the study cited in reference no. 35? Since they are derived from the same sample set, it is necessary to discuss the results and conclusions of both. Environmental samples, especially soils, can be difficult to standardize, and the type of soil directly influences the microbiota and microbiome of the sample.

There is an excessive amount of supplementary material, including different types of supplementary material: Supplementary Data, Supplementary Notes, Supplementary Figures, and Supplementary Tables. In 44 pages of supplementary material, there are 27 supplementary figures, many of them with several panels. Even the figures in the body of the text are difficult to interpret, as there are figures within figures, diagrams, tables. I believe that the authors need to reevaluate the structure of the article, making the text more dynamic, without having to stop reading every time to look for supplementary material. The article needs to be self-contained and the supplementary material used punctually to present a secondary result, not essential, for understanding the study.

The title is not very informative.

Line 47: update the reference

Line 69-70: This sentence sounds strange. Does the resistance gene confer higher fitness? Another point: in general, these MDR genes are limited to efflux pumps

Line 82: How to combat resistance genes in natural forest soils?

Line 87: what does AP mean?

Line 98-99: Check the numbers. They are different from the abstract and the introduction. Despite the large initial sample

size, the analyses needed to be limited to the smallest number of samples from each environment and randomized 999x. How did this downsampling affect the final result? What is the relevance of randomization for this study and for the p-value found in each analysis?

Line 111-112: The separation does not seem so clear to me. It is difficult to analyze whether there are samples behind the red dots of the "soil" sample. In addition, there are other more obvious separations. Observing table S4, all samples differ from each other when compared in pairs.

All analyses in the article use the ARG copy per cell metric, also proposed by the group in 2023 in the article doi: 10.1021/acs.est.3c00159. What would be the advantage of this metric over ARGs/genome?

Line 124-126: "The patterns of change in the relative abundance of total ARGs and Rank I ARGs were assessed under consistent data volume (Fig. S5a and b) and continental source (Fig. S5c)." It is unclear what this means.

Line 134-156 and figure 1g: I think there is a very important point here that needs to be carefully evaluated. The authors suggest that the soil receives ARGs from various sources, mainly human and animal feces. In general, human waste ends up going into the sewer and contaminating the water. The same occurs with animal waste, whose water used to wash the breeding sites ends up, after treatment, flowing into a river. In many cases, and this will depend on each country, river water is used to irrigate crops, which contaminates the soil and vegetables. Still according to figure 1g, the ARGs travel from the soil to some body of water. The influence of bodies of water was evaluated in the article in reference 35, hence the need for comparison. The way it was presented, I see the soil as an intermediary, the water as a disseminator and the feces as the great reservoirs of Rank 1 resistance genes. What would be the selective pressure for a bacterium to maintain a Rank 1 ARG in the soil?

Line 180: Is this statement based solely on Figure 2e or is there a study that can be referenced? If it is based on Figure 2e, I would add "our results suggest that..."

What is the definition of the term "Richness" in the context of this study?

Line 217-219: The analysis over time is very interesting!

Line 312: Reference or link to the databases.

Line 353-369: Considering the statements: "Considering that high antibiotic pressures are indeed uncommon in soil, even in agricultural soils" and "both the metagenomic data and E. coli isolate genomes confirmed that humans and livestock were the main sources of Rank I ARGs in soil", is it possible to interpret that livestock and human feces are the reservoirs of resistance genes, distributed by water currents?

Line 454: Were there any inclusion and exclusion criteria for the soil samples? Are they forest soils? If so, what type of forest? Vegetation is an important factor in the diversity of the soil microbiome and, consequently, in the gene content. Are they cerrado soils, highland soils, clayey soils, sandy soils, humus soils, or calcareous soils?

Line 457: Add criterion (iv). The term GB refers to Giga Byte. If it is Giga bases, the correct term is 1 Gb or 1 Gbp. Check.

Line 458-459: It is necessary to identify what are "potentially contaminated environments" since this is quite relevant for the interpretation of the data. What are the criteria? Contaminated with what?

Line 511: sequence mapping?

Line 553: ClustalW

Line 562: similarity or identity? Are you comparing DNA sequences?

Line 559-600: Based on experimental data? Broth dilution, diffusion disk, vitek?

Line 602-603: "we were focused on and only bacteria with a total number greater than 30 were included". To improve

Reviewer #2

(Remarks to the Author)

In this manuscript entitled "Deciphering the increasing risk of soil resistome associated with human resistance under One Health framework" the authors have analysed an impressive set of soil metagenomes for antibiotic resistance genes and tried to decipher the relationships and transmission pathways between soil and other environments. My main concern is the annotation of ARGs in this study, which lies the foundation for all further analysis. The authors have used their own pipeline, ARG-OAP, which includes several genes that have little to nothing to do with antibiotic resistance. Many of these are genes where a mutation causes the resistance phenotype or regulatory genes that per se do not confer resistance, but might be involved in giving a resistance phenotype. As this is a metagenomic study, detecting regulatory genes does not indicate

resistance and annotation of point mutations would need more in-depth analysis than just similarity searches, as has been done in this manuscript. Another key element in this paper is the resistance risk ranking, where the Rank 1 ARGs are considered to be of highest concern. This is an important aspect when studying antibiotic resistomes, but I have doubts of these rankings as well. The authors report Rank I ARGs such as *metE*, *mdtL*, *garX* and *catI*, from which only *catI* could be considered as ARG. The rest are *E. coli* core genes, found even from *E. coli* K12, hardly a dangerous pathogen. So I have doubts about these genes being ARGs of highest concern. Also the latest version of SARG database (<https://smile.hku.hk/ARGs/Indexing/riskranking>) lists only *catI* as Rank I ARG. As some of the authors of this manuscript are also behind the database, one would expect that correct rankings would have been used.

The authors have assembled the metagenomes in order to connect the ARGs to larger context. However, it is well known that especially mobile ARGs are very difficult to assemble to larger context from short-read data. This can be seen in the most abundant genes reported, mainly chromosomal genes from *E. coli*. So making conclusions based on these does not give a full picture of the antibiotic resistome.

The authors have also done a lot of statistics based on the p-values and r-values, but there is no explanation what was tested and how. Also it is not clear where the soil metadata was collected, or how can you measure "climate change" and use it as a variable in the models.

In addition, the accession number to the metagenomic data produced in this work (PRJNA877822) points to amplicon sequencing data from composts.

As most of the findings in this manuscript are based on what I consider false positives, or just bad annotations, I can only suggest rejection of this manuscript, because I do not believe that these results are correct. I however, encourage the authors to redesign the analysis, because I do think this is an important topic and the dataset collected is of value.

Version 1:

Reviewer comments:

Reviewer #1

(Remarks to the Author)

All my questions were answered satisfactorily and contributed to improving general understanding, however reading is still challenging.

Just a question about the mobility of ARGs raised by reviewer 2. If I understand correctly, according to the 2021 paper (<https://doi.org/10.1038/s41467-021-25096-3>), mobility is one of the conditions for an ARG to be classified as Rank I. In the revised version, the authors removed the analysis of ARGs from metagenomes and the discussion about their mobility. Only pure *E. coli* genomes were used. What are we seeing in figures 1b, 1d and 1f?

"We did not use assembled metagenomic contigs to discuss the mobility of ARGs. In our revised manuscript, we used the genomes of *E. coli* pure cultures to discuss potential mobility."

Reviewer #2

(Remarks to the Author)

The authors have addressed all of my comments and modified the manuscript accordingly. I have no further comments.

Reviewer #1 (Remarks to the Author):

In the paper entitled “Deciphering the increasing risk of soil resistome associated with human resistance under One Health framework”, the authors perform an extensive analysis of antimicrobial resistance genes in more than 3,000 metagenome samples and use approximately 8,000 E. coli genomes from different environments to link the environment, animals, and humans. The authors use a rank metric to classify the resistance gene as high risk (Rank 1), intermediate, or low. This metric was developed and published by the group in 2021 in the paper doi:10.1038/s41467-021-25096-3. As their main conclusion, the authors state that soil is the main reservoir of Rank 1 ARGs that reach humans. This conclusion is based on a metric developed by the authors that they called “connectedness”. All this information, previous and new, made reading, understanding, and interpreting this paper challenging. Below, I leave my main comments regarding the manuscript.

As I anticipated, reading, understanding, and interpreting this study was challenging. There are many concepts that need to be understood in order to correctly interpret the results presented in this manuscript. The description of the public metagenome samples is referenced in paper no. 35 (Yin, X. et al. Global environmental resistome: Distinction and connectivity across diverse habitats benchmarked by metagenomic analyses. Water Res. 235, 2023.), but the sample numbers are different, which made it necessary to look at the supplementary material of this other paper. What is the sampling relationship between this study and the study cited in reference no. 35? Since they are derived from the same sample set, it is necessary to discuss the results and conclusions of both.

Response: Thanks for these comments. According to your comments and suggestions, we have carefully revised the manuscript. The response text including: a. Black italic type: the exact comment. b. in normal font: response to the comment. c. in blue: revisions in manuscript. Below we provide our point-by-point responses to your comments and hope the revised manuscript will meet with your approval.

The dataset in our manuscript is different from that of Yin, X. et al. 2023. The soil dataset used in this study is independent and consists of a total of 2,540 soil metagenomic data (2,391 public data and 149 in-house data). Only the other habitats (excluding soil) metagenomic data refers to the 1,303 non-soil metagenomic samples with clearly defined habitat information collected by Yin, X. et al. (2023) (excluding soil). Additionally, considering the potential association between crop surface microbiome and soil, we also collected 122 metagenomic samples from crop surfaces, which are not included in the dataset of Yin, X. et al., 2023. To present our dataset more clearly, we divided it into Supplementary Data 1 and Supplementary Data 2 to distinguish between soil and other habitats (non-soil) metagenomic data and their corresponding SRA number. Additionally, we added some sentences to further clarify the collection of these data. We also added some discussions.

Line 396-400: “Yin et al. (2023) suggested that rivers with high human density, advanced economic development, and intensive industrial activities are more vulnerable to human activities, such as sewage plant discharge, than those in less developed regions (by more than five times). Thus, these results implied that soil might act as an intermediary, the water as a disseminator.”

Line 467-468: “The soil dataset contained 2,540 samples, of which 2,391 samples were public data and 149 were in-house data (Supplementary Data 1).”

Line 477-484: “We also included 1,303 metagenomic samples, collected from different habitats, such as swine faeces, cattle faeces, chicken faeces, human faeces, water from wastewater treatment plants, sewage, natural sediment, natural water, marine water, and landfill. The details of these samples were as previously described³⁶. Additionally, 122 crop surface metagenomic data downloaded. In summary, 1,425 metagenomic data, collected from other habitats, were analyzed. All data collected from other habitats are shown in Supplementary Data 2.”

Environmental samples, especially soils, can be difficult to standardize, and the type of soil directly influences the microbiota and microbiome of the sample.

Response: Thanks for this comment. Indeed, the type of soil directly influences the microbiome. Therefore, ten land-use types were included in this analysis. The rarefaction curves and profiles of ARGs across different land-use types were added in Fig. S4 and Fig. S6. To confirm the consistency of temporal trends, the 999 rounds of sampling based on the specified landuse combination were further performed (Fig. S9d). The results showed that despite the significant differences in ARGs composition among different land-use types, the overall trend remains consistent after subsampling (i.e., the relative abundance of Rank I ARGs showed an increasing trend over time). We added some sentences.

Line 108-110: “The profiles of antibiotic resistome in different habitats and different

landuse types (including rarefaction curves and composition of ARGs at the type and subtype level) are detailed in Fig. S3-S6.”

Line131-134: “The changes in the relative abundances of total ARGs and Rank I ARGs were assessed under consistent data volume (Fig. S9b), consistent continental origin combinations (Fig. S9c), and consistent land use type combinations (Fig. S9d).”

Line 468-470: “Land use and soil type for each sample were collected via ArcGis (v10.8) and were shown in Supplementary Data 1.”

Fig. S1 Soil rarefaction curves grouped by different land use type. (a) Type. (b) Subtype. (c) Gene.

Fig. S2 Composition of Total ARGs and Rank I ARGs in different landuse type. (a) Composition of Total ARGs in different landuse type (type level); (b) Composition of Total ARGs in different landuse type (subtype level); (c) Composition of Rank I ARGs in different landuse type (Subtype level).

Fig. S3 The relative abundance of Total ARGs and Rank I ARGs in different period. (a)

Each period uses all of the data. The red dots indicate the means. (b) Perform 999 rounds of sampling based on the minimum sample size in the period (Period A, 133 samples). (c) Perform 999 rounds of sampling based on the specified continent combination (15 from Europe, 15 from America, 10 from Asia, and 10 from others). (d) Perform 999 rounds of sampling based on the specified landuse combination (70 from farmland, 20 from forest, 10 from grassland, and 20 from others).

There is an excessive amount of supplementary material, including different types of supplementary material: Supplementary Data, Supplementary Notes, Supplementary Figures, and Supplementary Tables. In 44 pages of supplementary material, there are 27 supplementary figures, many of them with several panels. Even the figures in the body of the text are difficult to interpret, as there are figures within figures, diagrams, tables. I believe that the authors need to reevaluate the structure of the article, making the text more dynamic, without having to stop reading every time to look for supplementary material. The article needs to be self-contained and the supplementary material used punctually to present a secondary result, not essential, for understanding the study.

Response: Thanks for this comment. We streamlined and reconstructed the supplementary materials and moved some lengthy tables to the Supplementary Data. After adding 4 new supplemental figures, the total number of images is now 19, and the document spans 27 pages. The manuscript structure has been adjusted to make it more cohesive and fluid, improving the overall readability of the article. Key results and analyses are fully explained within the text, reducing the reliance on the supplementary materials. More detail legend was added to make the figure clearer. Additionally, we revised the results section to clarify why and how the analyses were conducted. Please refer to Section Supplementary material.

The title is not very informative.

Response: Thanks for this comment. The title was revised to “The Increasing Risk of

Soil Resistome and Its Implications for Human Antibiotic Resistance under One Health Framework”

Line 47: update the reference

Response: Thanks for this suggestion. The reference was updated.

Line 821-822: “Geneva, W. H. O. Global antimicrobial resistance and use surveillance system (GLASS) report; 2022.”

Line 69-70: This sentence sounds strange. Does the resistance gene confer higher fitness? Another point: in general, these MDR genes are limited to efflux pumps

Response: Thanks for this comment. The sentence was revised.

Line 70-71: “Multidrug efflux pumps are one of the main types of ARGs in soil¹⁴.”

Line 82: How to combat resistance genes in natural forest soils?

Response: Thanks for this comment. Although natural forest soils are not major reservoirs of Rank I ARGs, preventive measures, such as controlling pollution, and monitoring Rank I ARG, are the best approaches to minimizing resistance gene proliferation in these ecosystems.

Line 87: what does AP mean?

Response: Thanks for this comment. AP indicates ARGs carrying pathogen.

Line 87-90: “Meanwhile, we collected 8,388 *E. coli* genomes ((the main ARGs carrying

pathogen in soil) isolated from soil, livestock, and humans (main source), to reveal the potential exchange network of soil and other habitats Rank I ARGs at the species level.”

Line 98-99: Check the numbers. They are different from the abstract and the introduction.

Response: Thanks for this suggestions. A total of 3,966 metagenomic data were included in the analysis, including 2540 soil samples and 1426 other habitat samples. Among the 2,540 soil samples, 2,391 are public data, and 149 are in-house data. We revised the sentences.

Line 26-29: “To fill this gap, we analyzed 3,966 metagenomic data (including 11 habitats) and 8,388 genomes of *Escherichia coli* isolates (the main pathogen indicator in soil) in the public databases together with some in-house metagenomic samples (149 soil samples).”

Line 84-87: “Herein, we constructed a dataset that includes 3,817 metagenomic datasets (including 2,391 soil samples and 1,426 other habitat samples) from public databases and 149 in-house data to catalog the profile and attribution of the soil antibiotic resistome (i.e., 3,966 samples in summary).”

Despite the large initial sample size, the analyses needed to be limited to the smallest number of samples from each environment and randomized 999x. How did this downsampling affect the final result?

Response: Thanks for this suggestions. We performed 999 rounds of sampling based

on the minimum sample size in the habitat (Cattle_feces, 12 samples, Fig. S7). The results were consistent with the analysis without normalization. We added the sentence. Line 114-115: “The similar trends could also be observed after the minimum sample size for each habitat is standardized (Fig. S7).”

Fig. S4 The relative abundance of Total ARGs and Rank I ARGs in different habitats.

- (a) Total ARGs. In the boxplots of panels, hinges indicate the 25th, 50th, and 75th percentiles, whiskers indicate $1.5 \times$ interquartile ranges, dots indicate the average value from each random sampling. Perform 999 rounds of sampling based on the minimum sample size in the habitat (Cattle_feces, 12 samples).
- (b) Rank I ARGs. In the boxplots of panels, hinges indicate the 25th, 50th, and 75th percentiles, whiskers indicate $1.5 \times$ interquartile ranges, dots indicate the average value from each random sampling. Perform 999 rounds of sampling based on the minimum sample size in the habitat (Cattle_feces, 12 samples).

What is the relevance of randomization for this study and for the p-value found in each analysis?

Response: Thanks for this suggestions. Randomization has little impact on the study's

conclusions and does not change the trend of the p-value. The randomization test on the years were also conducted and found that the trend and p-value remained consistent. We added the analysis on both metagenomic data and *E. coli* isolate genome and sentence.

Line 130-136: “To further ensure the reliability of the trends, we performed data normalization on both year (Fig. S8) and period (Fig. S9a). The changes in the relative abundances of total ARGs and Rank I ARGs were assessed under consistent data volume (Fig. S9b), consistent continental origin combinations (Fig. S9c), and consistent land use type combinations (Fig. S9d). All these results were consistent with previous findings, indicating that the relative abundance of total ARGs was time-independent, but ARGs risk showed a significant increase over time.”

Line 204-205: “Even after rarefying based on the same sample size, the consistent trend remains (Fig. S13).”

Fig. S5 The relative abundance of Total ARGs and RankI ARGs in different Year. (a) Total ARGs. Perform 999 rounds of sampling based on the minimum sample size in each year (2008, 24 samples). (b) RankI ARGs. Perform 999 rounds of sampling based on the minimum sample size in each year (2008, 24 samples).

Fig. S6 The relative abundance of RankI ARGs and richness in different Year. (a) Mean copy number of RankI ARGs. Perform 999 rounds of sampling based on the 10 samples in each year. If samples in this year fewer than 10, all will be included in the analysis. (b) Mean copy number of MRankI ARGs. (c) Richness of *E. coli* Rank I ARGs (per genome). (d) Richness of *E. coli* MRank I ARGs (per genome). Pearson correlation (loess regression) was conducted to obtained the *p* value and *r*.

Line 111-112: The separation does not seem so clear to me. It is difficult to analyze whether there are samples behind the red dots of the “soil” sample. In addition, there are other more obvious separations. Observing table S4, all samples differ from each other when compared in pairs.

Response: Thanks for this comment. To make the difference more clear, we uploaded

the 3D versions “3D_PCoA_Total.html” and “3D_PCoA_RankI.html” to highlight the differences among different habitats. We added sentence and links.

Line 676-678: “A version with multiple viewing angles is available at https://github.com/Yuxiang-Zhao/ARGs/blob/main/PCoA/3D_PCoA_Total.html.”

Line 680-680: “A version with multiple viewing angles is available at https://github.com/Yuxiang-Zhao/ARGs/blob/main/PCoA/3D_PCoA_RankI.html.”

All analyses in the article use the ARG copy per cell metric, also proposed by the group in 2023 in the article doi: 10.1021/acs.est.3c00159. What would be the advantage of this metric over ARGs/genome?

Response: Thanks for this comment. In the metagenomic analysis, the advantage of the unit “ARG copy per cell” is that it can be transformed to absolute abundance data with ease if cell abundance data are available or scaling factors based on cell spike-ins can be applied. This unit can be further developed into absolute quantification of viable and membrane-compromised dead antibiotic resistant bacteria cells if combined with the pretreatment using a cell-impermeant dye like propidium monoazide. When assuming one genome per cell, the ARG copy number per cell is equivalent to the copy number per genome. Therefore, in the analysis of isolate genomes, ARG copy per cell is equivalent to ARG copy per genome.

Line 124-126: “The patterns of change in the relative abundance of total ARGs and Rank I ARGs were assessed under consistent data volume (Fig. S5a and b) and continental source (Fig. S5c).” It is unclear what this means.

Response: Thanks for this comment. We revised the sentence.

Line 131-134: “The changes in the relative abundances of total ARGs and Rank I ARGs were assessed under consistent data volume (Fig. S9b), consistent continental origin combinations (Fig. S9c), and consistent land use type combinations (Fig. S9d).”

Line 134-156 and figure 1g: I think there is a very important point here that needs to be carefully evaluated. The authors suggest that the soil receives ARGs from various sources, mainly human and animal feces. In general, human waste ends up going into the sewer and contaminating the water. The same occurs with animal waste, whose water used to wash the breeding sites ends up, after treatment, flowing into a river. In many cases, and this will depend on each country, river water is used to irrigate crops, which contaminates the soil and vegetables. Still according to figure 1g, the ARGs travel from the soil to some body of water. The influence of bodies of water was evaluated in the article in reference 35, hence the need for comparison. The way it was presented, I see the soil as an intermediary, the water as a disseminator and the feces as the great reservoirs of Rank I resistance genes.

Response: Thanks for this suggestion. The arrows in Fig. 1 do not indicate the actual transmission direction but rather represent a hypothesized flow based on the assumption that ARGs spread from high to low concentrations. To avoid misunderstandings, we have removed the arrows from Fig. 1g. Indeed, since river water is used to irrigate crops, water flow could be one of the important pathways for ARGs to spread into the soil. Our previous study suggested that the downstream of the Yangtze River (characterized

by high human density, advanced economic development, and intensive industrial activities) is more than five times as vulnerable to human activities to the upstream. Thus, water might serve as a disseminator, and feces act as the great reservoirs of Rank 1 resistance genes. We added more discussion.

Line 393-400: “Based on potential attribution analysis, feces were identified as the great reservoirs of Rank I ARGs (Fig. 1g). In general, human and animal waste ultimately enters the sewer system, leading to water contamination. Yin et al. (2023) suggested that rivers with high human density, advanced economic development, and intensive industrial activities are more vulnerable to human activities, such as sewage plant discharge, than those in less developed regions (by more than five times). Thus, these results implied that soil might act as an intermediary, the water as a disseminator.”

What would be the selective pressure for a bacterium to maintain a Rank 1 ARG in the soil?

Response: Thanks for this comment. The selective pressure for a bacterium to maintain a Rank 1 antibiotic resistance gene (ARG) in the soil arises from multiple environmental factors, including 1) continuous introduction of bacteria carrying Rank 1 ARGs, 2) horizontal gene transfer driven by various environmental stress, 3) trace antibiotic residues, and 4) co-selection caused by heavy metal and other pollutants. We added sentences in the discussion.

Line 355-358: “Continuous introduction of bacteria carrying Rank 1 ARGs, HGT driven by environmental stress, trace antibiotic residues, and co-selection caused by

various pollutants might be the potential mechanisms for the existence of Rank 1 ARGs in soil.”

Line 180: Is this statement based solely on Figure 2e or is there a study that can be referenced? If it is based on Figure 2e, I would add “our results suggest that...”

Response: Thanks for this suggestion. We revised the sentence.

Line 178-180: “Our results suggested that *E. coli* was the most abundant pathogen carrying ARGs on chromosomes, accounting for 46.0%, followed by a plant pathogen *Pectobacterium carotovorum* (27.2%) (Fig. 2e).”

What is the definition of the term “Richness” in the context of this study?

Response: Thanks for this comment. Richness refers to the number of ARGs types, subtypes, and gene types present. We added the definition.

Line 110-114: “Briefly, the richness (number of subtype) and relative abundance of total ARGs (1739 subtypes, 0.18 copies per cell) and Rank I ARGs (175 gene types, 0.3 copies per 1000 cell) in soil were similar to those in wastewater treatment plant (WWTP) samples but lower than those in livestock and human faeces (Fig. 1a and b).”

Line 199-201: “We focused on the temporal trends in the mean copy number of Rank I and MRank I ARGs, the occurrence frequency of MRank I ARGs, and their richness (number of Rank I / MRank I gene types) in soil source *E. coli*. (Fig. 3a and b).”

Line 729-730: “Richness refers to the number of Rank I and MRank I ARGs gene type.”

Line 217-219: The analysis over time is very interesting!

Response: Thanks for this kind comment.

Line 312: Reference or link to the databases.

Response: Thanks for this suggestion. We added the link to these databases.

Line 625-632: “For this global analysis, we collected the human clinical antibiotic resistance genes from 5 sources, including Resistancemap (<https://resistancemap.onehealthtrust.org/>), European Centre for Disease Prevention and Control Surveillance Atlas (<https://www.ecdc.europa.eu/en/surveillance-atlas-infectious-diseases>), PLISA Health Information Platform for the Americas (<https://opendata.paho.org/en>), World Health Organization (<https://dev-cms.who.int/initiatives/glass>), and China Antimicrobial Resistance Monitoring System (<https://www.carss.cn/sys/Htmls/dist/index.html>).”

Line 353-369: Considering the statements: “Considering that high antibiotic pressures are indeed uncommon in soil, even in agricultural soils” and “both the metagenomic data and E. coli isolate genomes confirmed that humans and livestock were the main sources of Rank I ARGs in soil”, is it possible to interpret that livestock and human feces are the reservoirs of resistance genes, distributed by water currents?

Response: Thanks for this suggestion. As you mentioned, water plays a crucial role in the dissemination of ARGs. Please refer to previous response for further details.

Line 454: Were there any inclusion and exclusion criteria for the soil samples? Are they forest soils? If so, what type of forest? Vegetation is an important factor in the diversity of the soil microbiome and, consequently, in the gene content. Are they cerrado soils, highland soils, clayey soils, sandy soils, humus soils, or calcareous soils?

Response: Thanks for this comment. We have inclusion criteria for the data, including: (i) Illumina shotgun data; (ii) paired-end data with FASTQ format; (iii) over 1 Giga Byte (GB); (iv) no culturing or any other additional experiments; (v) not collected from potentially contaminated environments, including heavy metal contamination, coking sites, industrial waste, etc.; (vi) included detailed sample information (e.g., accurate coordinate and sampling time); (vii) average read length over 100 base pairs. Our dataset includes not only forests but a total of 10 land use types, primarily including forests, grasslands, and farmland. As your suggestion, we extracted detailed soil names (e.g., Humic Acrisols, Eutric Gleysols, Mollic Gleysols, etc.) corresponding to the sampling points from the Harmonized World Soil Database v2.0 (<https://www.fao.org/soils-portal/data-hub/soil-maps-and-databases/harmonized-world-soil-database-v20/zh/>) using ARCGIS 10.8. The results have been added to Supplementary Data 1. Indeed, vegetation is an important factor in the diversity of the soil microbiome. However, there is currently a lack of remote sensing information on detailed vegetation. In our future sampling efforts, we will focus on and record vegetation in detail. Additionally, we added sentence.

Line 361-364: “Considering that high antibiotic pressures are indeed uncommon in soil, even in agricultural soils³⁸, it is widely recognized that the soil antibiotic resistome is

influenced by environmental factors (e.g., pH¹⁴, total organic carbon³⁹), location (i.e., longitude and latitude)¹², vegetation⁴⁰, and climatic seasonality (i.e., cold weather)¹¹.”

Line 468-470: “Land use and soil type for each sample were collected via ArcGis (v10.8) and were shown in Supplementary Data 1.”

Line 454-457: “Due to the lack of detailed vegetation, it was unable to fully consider vegetation factors. Future research could focus on and document the vegetation status at sampling sites to deeply explore the impact of vegetation on soil ARGs resistome.”

Line 457: Add criterion (iv). The term GB refers to Giga Byte. If it is Giga bases, the correct term is 1 Gb or 1 Gbp. Check.

Response: Thanks for this comment. We corrected the numbering and changed "GB" to "Giga Byte."

Line 472: “(iii) over 1 Giga Byte (GB)”

Line 458-459: It is necessary to identify what are “potentially contaminated environments” since this is quite relevant for the interpretation of the data. What are the criteria? Contaminated with what?

Response: Thanks for this comment. Contaminated soil refers to soil polluted by heavy metals, organic pollutants, or industrial waste. We revised the sentence.

Line 473-474: “(v) not collected from potentially contaminated environments, including heavy metal contamination, coking sites, industrial waste, etc.;;”

Line 511: sequence mapping?

Response: Thanks for this suggestion. We revised the sentence.

Line 540-542: “Bowtie2 (version 2.4.2) was used for sequence mapping (very-sensitive), and Samtools (version 1.11) was used to process and convert the alignment results.”

Line 553: ClustalW

Response: Thanks for this suggestion. We revised the sentence.

Line 582-584: “The sequences were aligned using ClustalW (v2.1) and phylogenetic trees were constructed with FastTree (v2.1).”

Line 562: similarity or identity? Are you comparing DNA sequences?

Response: Thanks for this comment. We compared DNA sequence and used identity.

We revised the sentence.

Line 592-593: “Only DNA sequences with greater than 100% identity, and those that translate into complete ORFs, were retained.”

Line 559-600: Based on experimental data? Broth dilution, diffusion disk, vitek?

Response: Thanks for this comment. These analysis were based on the collect *E.coli* isolate genome. We added sentence to clarify it.

Line 591-592: “Blastn (v2.6.0) was further used to screen blocks of DNA that were shared by two collected *E.coli* isolate genomes.”

Line 602-603: “we were focused on and only bacteria with a total number greater than 30 were included”. To improve

Response: Thanks for this suggestion. We revised the sentence.

Line 637-638: “**Only bacteria with a total number greater than 30 were included in the analysis.**”

Reviewer #2 (Remarks to the Author):

In this manuscript entitled "Deciphering the increasing risk of soil resistome associated with human resistance under One Health framework" the authors have analysed an impressive set of soil metagenomes for antibiotic resistance genes and tried to decipher the relationships and transmission pathways between soil and other environments. My main concern is the annotation of ARGs in this study, which lies the foundation for all further analysis. The authors have used their own pipeline, ARG-OAP, which includes several genes that have little to nothing to do with antibiotic resistance. Many of these are genes where a mutation causes the resistance phenotype or regulatory genes that per se do not confer resistance, but might be involved in giving a resistance phenotype. As this is a metagenomic study, detecting regulatory genes does not indicate resistance and annotation of point mutations would need more in-depth analysis than just similarity searches, as has been done in this manuscript.

Response: Thanks for these comments. According to your comments and suggestions, we have carefully revised the manuscript. To address your concerns, we conducted the following steps: (1) Reanalysis using the SARG3.0_S database, which excluded regulators and mutations; (2) Reanalysis with the latest version of Rank I ARGs (2023 version); (3) Exclusion of the multidrug efflux pump related genes; (4) Revision of all the figures, tables, and related sentences. The response text including: a. Black italic type: the exact comment. b. in normal font: response to the comment. c. in blue: revisions in manuscript. Below we provide our point-by-point responses to your comments and hope the revised manuscript will meet with your approval.

Following the comments of Reviewers, in this revised manuscript, we replaced the database SARG3.0_F with a new version (SARG3.0_S) and **reanalyzed all the figures, tables and related sentences in both the manuscript and supplementary materials.** SARG3.0_S is a database derived from SARG3.0_F by removing 926 sequences tagged with transcriptional regulators (including activators and repressors), point mutations, and others¹, for similarity search annotation (Fig. R1). In addition, considering the controversy surrounding multidrug efflux pumps, all genes related to these pumps were excluded in the revised manuscript to avoid possible mis-annotations of ARGs. **Overall, the results showed that although the detailed values changed, the trends remained unchanged, supporting the reliability of our conclusions.** Once again, we sincerely appreciate your valuable feedback on our work. Please refer to manuscript and supplementary materials.

Tag	SARG. Seq. I	Type	Subtype	HMM. categ	Mechanism	Mechanism	Mechanism	SARG3.0_F	SARG3.0_S
mutation	gb AA04722	aminoc	aminocour	Strepto	mutation			Yes	No
overexpression	gb AAC7378	aminogl	aminoglyc	dpE	overexpression			Yes	No
overexpression	AEB80130	bacitra	bacitracib	crC	overexpression			Yes	No
overexpression	F42334	bacitra	bacitracib	crC	overexpression			Yes	No
regulator	EF540343.1	beta_la	beta_lactbla	R1	regulator			Yes	No
regulator	gi 1011940	beta_la	beta_lactbla	R1	regulator			Yes	No
regulator	gi 1027664	beta_la	beta_lactbla	R1	regulator			Yes	No
regulator	gi 7372177	beta_la	beta_lactbla	R1	regulator			Yes	No
regulator	gi 9220635	beta_la	beta_lactbla	R1	regulator			Yes	No
repressor	CP000675.2	beta_la	beta_lactnae	I	repressor			Yes	No
repressor	FR823292.1	beta_la	beta_lactnae	I	repressor			Yes	No
repressor	gi 1005816	beta_la	beta_lactnae	I	repressor			Yes	No
repressor	gi 1029888	beta_la	beta_lactnae	I	repressor			Yes	No
repressor	gi 1031792	beta_la	beta_lactnae	I	repressor			Yes	No
repressor	gi 1031805	beta_la	beta_lactnae	I	repressor			Yes	No
repressor	gi 1035887	beta_la	beta_lactnae	I	repressor			Yes	No
repressor	gi 1041152	beta_la	beta_lactnae	I	repressor			Yes	No
repressor	gi 2791921	beta_la	beta_lactnae	I	repressor			Yes	No
repressor	gi 2885517	beta_la	beta_lactnae	I	repressor			Yes	No
repressor	gi 4139155	beta_la	beta_lactnae	I	repressor			Yes	No
repressor	gi 4462913	beta_la	beta_lactnae	I	repressor			Yes	No
repressor	gi 4462913	beta_la	beta_lactnae	I	repressor			Yes	No
repressor	gi 4462913	beta_la	beta_lactnae	I	repressor			Yes	No
repressor	gi 4601576	beta_la	beta_lactnae	I	repressor			Yes	No
repressor	gi 4884164	beta_la	beta_lactnae	I	repressor			Yes	No
repressor	gi 5565028	beta_la	beta_lactnae	I	repressor			Yes	No
repressor	gi 5822224	beta_la	beta_lactnae	I	repressor			Yes	No
repressor	gi 6039264	beta_la	beta_lactnae	I	repressor			Yes	No
repressor	gi 6281594	beta_la	beta_lactnae	I	repressor			Yes	No
repressor	gi 6476311	beta_la	beta_lactnae	I	repressor			Yes	No
repressor	gi 6864342	beta_la	beta_lactnae	I	repressor			Yes	No
repressor	gi 7277441	beta_la	beta_lactnae	I	repressor			Yes	No
repressor	gi 8274544	beta_la	beta_lactnae	I	repressor			Yes	No
repressor	gi 8963560	beta_la	beta_lactnae	I	repressor			Yes	No
repressor	gi 8965331	beta_la	beta_lactnae	I	repressor			Yes	No
repressor	gi 9229799	beta_la	beta_lactnae	I	repressor			Yes	No
repressor	gi 9956155	beta_la	beta_lactnae	I	repressor			Yes	No
repressor	NC_002745.	beta_la	beta_lactnae	I	repressor			Yes	No
repressor	NC_002758.	beta_la	beta_lactnae	I	repressor			Yes	No
repressor	NC_002952.	beta_la	beta_lactnae	I	repressor			Yes	No
regulator	EAC57479	beta_la	beta_lactnae	R1	regulator			Yes	No
regulator	FAC2437R	beta_la	beta_lactnae	R1	regulator			Yes	No

Fig. R1 The structure of SARG3.0_S compared to SARG3.0_F. Mutation, overexpression, regulator, etc. were labeled in the Tag list. The last two

columns show whether an ARG is contained in a database. Please refer to https://github.com/Yuxiang-Zhao/ARGs/tree/main/database_structure.

Another key element in this paper is the resistance risk ranking, where the Rank I ARGs are considered to be of highest concern. This is an important aspect when studying antibiotic resistomes, but I have doubts of these rankings as well. The authors report Rank I ARGs such as metE, mdtL, garX and catI, from which only catI could be considered as ARG. The rest are E. coli core genes, found even from E. coli K12, hardly a dangerous pathogen. So I have doubts about these genes being ARGs of highest concern. Also the latest version of SARG database (<https://smile.hku.hk/ARGs/Indexing/riskranking>) lists only catI as Rank I ARG. As some of the authors of this manuscript are also behind the database, one would expect that correct rankings would have been used.

Response: Thanks for this comment. As your suggestion, we reanalyzed the data using the latest version of Rank I ARGs (<https://smile.hku.hk/ARGs/Indexing/riskranking>, 2023 version). A total of 214 Rank I ARGs were included in the revised manuscript (Supplementary Data 3), and those genes pointed out by the Reviewer was not included in these 214 Rank I ARGs. **We revised all analyses involving Rank I ARGs. Overall, the results showed that although the detailed values changed, the trends remained unchanged, supporting the reliability of our conclusions.**

The authors have assembled the metagenomes in order to connect the ARGs to larger context. However, it is well known that especially mobile ARGs are very difficult to

assemble to larger context from short-read data. This can be seen in the most abundant genes reported, mainly chromosomal genes from E. coli. So making conclusions based on these does not give a full picture of the antibiotic resistome.

Response: Thanks for this comment and we agree with the Reviewer that making conclusions based on contigs assembled from metagenomic data sets does not give a full picture of the antibiotic resistome. We did not use assembled metagenomic contigs to discuss the mobility of ARGs. In our revised manuscript, we used the genomes of *E. coli* pure cultures to discuss potential mobility.

The authors have also done a lot of statistics based on the p-values and r-values, but there is no explanation what was tested and how.

Response: Thanks for this comment. Pearson correlation analysis and linear models were used, and the detailed methods and parameters have been provided in the related figure legend.

Also it is not clear where the soil metadata was collected, or how can you measure "climate change" and use it as a variable in the models.

Response: Thanks for this comment. The metadata was extracted from relevant climate maps (e.g., <https://worldclim.org/data/worldclim21.html>, <https://sage.nelson.wisc.edu/>) using ArcGIS based on the coordinates of the samples. According to Reviewers' comments, we have removed this section from the supplementary materials.

In addition, the accession number to the metagenomic data produced in this work (PRJNA877822) points to amplicon sequencing data from composts.

Response: Thanks for this comment. We revised the sentences and the accession numbers.

Line 640-647: “The in-house metagenomic sequencing data generated in this study have been deposited in the National Center for Biotechnology Information (NCBI) Sequence Read Archive (SRA) database under accession number PRJNA1202346 [<https://www.ncbi.nlm.nih.gov/bioproject/PRJNA1202346>] and PRJNA1229199 [<https://www.ncbi.nlm.nih.gov/bioproject/PRJNA1229199>]. The raw data will be made publicly available after the article is accepted. The temporal link used for review: <https://dataview.ncbi.nlm.nih.gov/object/PRJNA1202346?reviewer=ecolbvfgop3ccvq5fgno96ds8p>.”

As most of the findings in this manuscript are based on what I consider false positives, or just bad annotations, I can only suggest rejection of this manuscript, because I do not believe that these results are correct. I however, encourage the authors to redesign the analysis, because I do think this is an important topic and the dataset collected is of value.

Response: Thank you for your thoughtful and constructive feedback. We appreciate your great efforts on the comprehensive assessment of our manuscript. We agree with your concerns about false positives or bad annotations. Following your advices, we have carefully redesigned and re-conducted all the analyses to demonstrate the value of

the collected datasets on this important topic. We hope that the revisions can address your concerns and present an improved manuscript for your further review. Thank you again for your time and consideration.

Remaining reviewer comments:

Reviewer #1:

All my questions were answered satisfactorily and contributed to improving general understanding, however reading is still challenging.

*Just a question about the mobility of ARGs raised by reviewer 2. If I understand correctly, according to the 2021 paper (<https://doi.org/10.1038/s41467-021-25096-3>), mobility is one of the conditions for an ARG to be classified as Rank I. In the revised version, the authors removed the analysis of ARGs from metagenomes and the discussion about their mobility. Only pure *E. coli* genomes were used. What are we seeing in figures 1b, 1d and 1f?*

*"We did not use assembled metagenomic contigs to discuss the mobility of ARGs. In our revised manuscript, we used the genomes of *E. coli* pure cultures to discuss potential mobility."*

Response: Thanks for this comment. In our previous work^{1,2}, we established a list of 215 Rank I ARG variants, based on their association with pathogenicity, mobility, and human-associated enrichment. In this study, we analyzed ARGs profiles derived from 3,965 metagenomic samples, and compared them to the established Rank I ARGs list in our previous work^{1,2} to identify the presence of those specific variants within these samples. Therefore, the profiles in Figure 1 (panels b, d, f) represent the Rank I ARGs across the metagenomic samples, which were identified based on the list of Rank I ARGs in the previous reports². As described in the previous reports¹, the Rank I ARGs was identified as those ARGs enriched by anthropogenic activity, mobility and carried

by pathogens. The list is provided in Supplementary Data 3. The data in figure 1 panels b, d and f are about Rank I ARGs, so they are linked to mobility based the classification of Rank I ARGs. We added sentences to clarify.

Line 74-75: “Zhang et al. (2021) developed an “omics-based” framework to evaluate ARG risk and identified the list of Rank I ARGs²³.”

Line 101-109: “The profile of Rank I ARGs was obtained based on the previously defined list, characterized by reported host pathogenicity, gene mobility, and enrichment in human-associated environments²⁶. In detail, after obtained the ARGs profile for 3,965 metagenomic samples, we compared them to a previously established Rank I ARGs list to identify the presence of specific variants within these samples. The Rank I ARGs profiles represented the observation of these ARGs across the detected metagenomic samples in this study, while the classification of Rank I ARGs is based on the previous reports (Supplementary Data 3)²⁶.”

Lines 520-522: “To classify Rank I ARGs, we compared our data against a previously reported list (Supplementary Data 3)²⁶.”

Lines 839-840, Lines 849-850, and Lines 858-860: “The classification of Rank I ARGs was based on a comparison with the previously reported list (Supplementary Data 3)²⁶.”

Reference

1. Zhang, A. et al. An omics-based framework for assessing the health risk of antimicrobial resistance genes. *Nat. Commun.* 12, 4765 (2021).
2. Yin, X. et al. ARGs-OAP v3.0: antibiotic-resistance gene database curation and

analysis pipeline optimization. *Engineering*. 27, 234-241 (2023).

Reviewer #2:

The authors have addressed all of my comments and modified the manuscript accordingly. I have no further comments.

Response: Thanks for this comment.